# Distinct calcium sources regulate temporal profiles of NMDAR and mGluR-mediated protein synthesis

Sarayu Ramakrishna[1], Bindushree K Radhakrishna[1,2], Ahamed P Kaladiyil[1], Nisa Manzoor Shah[1,2], Nimisha Basavaraju[1,2], Kristine K Freude[3] (ORCID), Reddy Peera Kommaddi[1], Ravi S Muddashetty[1] (ORCID)

**Calcium signaling is integral for neuronal activity and synaptic plasticity. We demonstrate that the calcium response generated by different sources modulates neuronal activity–mediated protein synthesis, another process essential for synaptic plasticity. Stimulation of NMDARs generates a protein synthesis response involving three phases—increased translation inhibition, followed by a decrease in translation inhibition, and increased translation activation. We show that these phases are linked to NMDAR-mediated calcium response. Calcium influx through NMDARs elicits increased translation inhibition, which is necessary for the successive phases. Calcium through L-VGCCs acts as a switch from translation inhibition to the activation phase. NMDAR-mediated translation activation requires the contribution of L-VGCCs, RyRs, and SOCE. Furthermore, we show that IP3-mediated calcium release and SOCE are essential for mGluR-mediated translation up-regulation. Finally, we signify the relevance of our findings in the context of Alzheimer's disease. Using neurons derived from human fAD iPSCs and transgenic AD mice, we demonstrate the dysregulation of NMDAR-mediated calcium and translation response. Our study highlights the complex interplay between calcium signaling and protein synthesis, and its implications in neurodegeneration.**

## Introduction

Calcium, an important secondary messenger, lies at the hub of multiple signaling pathways (1). The increase in cytosolic calcium contributed by different calcium-permeable channels can create transient high-calcium microenvironments and activate distinct signaling cascades (1, 2). In neurons, the spatio-temporal calcium changes and calcium-mediated signaling pathways act as critical regulators of synaptic plasticity (2, 3). Although calcium release from the Endoplasmic Reticulum (ER) and store-operated calcium entry (SOCE) are common mechanisms in all cell types, neurons have additional contributions through Voltage-Gated Calcium Channels (VGCCs) and calcium-permeable ionotropic glutamate receptors such as NMDA receptors (NMDARs). Thus, in neurons, NMDARs, VGCCs, SOCE, and ER-mediated calcium release are the major routes contributing to local and global calcium elevations and activation of intracellular signaling pathways (2, 3, 4, 5). Many processes in neurons, such as gene expression, transcription, and alternative splicing, are shown to be regulated in a calcium-dependent and calcium channel–specific manner (6, 7, 8, 9, 10, 11, 12, 13, 14). Though few studies suggest that cytosolic calcium changes can direct protein synthesis response in non-neuronal cells (15, 16, 17, 18, 19), there is limited understanding of the link between calcium signals generated by specific channels and their effect on neuronal protein synthesis.

Neurons, being highly polarized cells, undergo immense spatio-temporal regulation of protein synthesis, which is essential for the maintenance of synaptic structure, function, and plasticity (20, 21, 22, 23). Regulation of protein synthesis is a common feature of different neuronal stimulation paradigms (e.g., glutamate, dopamine, BDNF), facilitating the activity-mediated changes in synaptic plasticity (22). Glutamate receptors are primary drivers of synaptic plasticity in the central nervous system, and among them, NMDARs and metabotropic glutamate receptors (mGluRs) are the major modulators of synaptic activity (24). Stimulation of NMDARs and mGluRs is known to generate specific spatio-temporal translation responses (25). Simultaneously, these receptors initiate unique calcium responses involving different channels (2, 3). Stimulation of NMDARs generates a specific calcium response initiated by calcium influx through NMDARs itself, followed by activation of L-VGCCs, Calcium Induced Calcium Release (CICR) from the ER, and SOCE to sustain the cytosolic calcium increase (5, 26, 27). Likewise, stimulation of mGluRs increases cytosolic calcium predominantly through IP3 receptors (IP3Rs) and SOCE (28, 29). In terms of translation, the stimulation of NMDARs results in a specific temporal pattern of protein synthesis, marked by an initial phase of translation inhibition followed by subsequent activation of translation (25, 30), whereas mGluR stimulation leads to robust translation activation response (25, 31, 32). Thus, distinct spatio-temporal calcium

[1]Centre for Brain Research, Indian Institute of Science, Bangalore, India [2]Manipal Academy of Higher Education, Manipal, India [3]Department of Veterinary and Animal Sciences, Faculty of Health and Medical Sciences, University of Copenhagen, Frederiksberg C, Denmark

Correspondence: ravimshetty@cbr-iisc.ac.in

responses involving multiple channels are generated on neuronal activity, which could act as primary orchestrators of the stimulation-specific protein synthesis profiles.

Exploration of the link between calcium signal and translation response on neuronal activity is not only important in the physiological condition, but it is also essential for understanding the molecular pathology of neurodegenerative diseases. Dysregulation of calcium homeostasis is a commonly observed cellular defect in many neurodegenerative diseases, including Alzheimer's disease (AD) (33, 34, 35, 36, 37). Presenilin (*PSEN*), one of the genes implicated in familial AD, is also reported to act as a calcium leak channel on the ER (38, 39, 40). In addition, several studies have established the impairment of neuronal protein synthesis in AD (41, 42, 43, 44, 45). In our prior study, we observed that neurons exposed to APOE4 (a genetic risk factor for Alzheimer's disease) exhibited suppression of both basal and NMDAR-mediated protein synthesis responses, attributed to the disruption of calcium homeostasis (30). Hence, the dysregulated calcium signal and defective protein synthesis in AD may be connected, potentially adding to the complexity of synaptic pathology in AD.

In this study, we show that calcium signaling is a critical regulator of activity-mediated protein synthesis in neurons. We investigate the contribution of distinct calcium sources in regulating specific phases of translation response on both NMDAR and mGluR stimulations. Finally, we highlight the relevance of our findings in the context of neurodegeneration by showing the defect in NMDAR-mediated calcium and translation response in AD neurons derived from fAD iPSCs and transgenic AD mice.

# Results

### NMDAR-mediated protein synthesis is regulated by calcium signals generated by distinct sources

Previously, we have reported that stimulation of NMDARs in primary cortical neurons generates a unique biphasic protein synthesis profile involving initial translation inhibition followed by a phase of translation activation (30). Stimulation of NMDARs is also known to generate a distinct spatio-temporal calcium response in neurons involving NMDARs, L-VGCCs, and calcium release from Ryanodine Receptors (RyRs) on the ER (5, 26, 27, 46, 47). To assess this further, rat primary cortical neurons (DIV15) were used for measuring the calcium response on NMDAR stimulation in the presence or absence of L-VGCC antagonist nifedipine and RyR antagonist dantrolene (Fig 1A). Blocking L-VGCCs using nifedipine significantly reduced the NMDAR-mediated calcium influx by close to 60% at the first, second, third, fourth, and fifth minute of NMDA addition (Fig 1B–G). Blocking RyRs using dantrolene reduced NMDAR-mediated calcium influx by 50% at the first and fifth minute after NMDA addition (Fig 1B–G), though the trend of decrease persisted along the second, third, and fourth minute time-points as well (Fig 1B–G). Thus, cytosolic calcium increase on NMDAR stimulation involves calcium influx from external sources (NMDARs and L-VGCCs), as well as calcium release from ER (RyR-mediated) with distinct temporal contributions.

To study the effect of different calcium sources on the translation response, we investigated the individual contribution of L-VGCCs and RyRs in the background of NMDAR stimulation. We stimulated the neurons with NMDA in the presence or absence of nifedipine or dantrolene, and measured the protein synthesis response using FUNCAT. In addition, to study the effect of IP3Rs and SOCE on NMDAR-mediated translation, we used 2-APB, which acts as a blocker for both IP3Rs and SOCE (Fig S1A). Pre-incubation of neurons with nifedipine, dantrolene, and 2-APB did not show a change in FUNCAT signal compared with the control condition, indicating that the drugs do not affect protein synthesis on their own (Fig S1B and C). Because we used methionine-free DMEM in the FUNCAT assay, we used puromycin incorporation as another readout to confirm that Met-free DMEM does not affect protein synthesis on its own. The change of media to Met-free DMEM or Neurobasal does not affect protein synthesis on its own compared with no medium change condition (Fig S1D). Thus, after these validations, we used FUNCAT as the primary readout for protein synthesis.

As reported previously, NMDAR stimulation for 5 min caused a decrease in the FUNCAT signal indicating translation inhibition (Fig 1H), which shifted to the phase of translation activation by 20 min, as observed by the increase in the FUNCAT signal (Fig 1H). In the presence of nifedipine (L-VGCC antagonist), we observed a decrease in FUNCAT signal at both 5- and 20-min time points of NMDAR stimulation compared with the basal condition (Fig 1I). Similarly, even in the presence of dantrolene (RyR antagonist), the FUNCAT signal was decreased at 5- and 20-min time points on NMDAR stimulation compared with the basal condition (Fig 1J). NMDAR stimulation in the presence of 2-APB caused a reduction in the FUNCAT signal at 5 min and no change in the FUNCAT signal at the 20-min time point compared with the basal condition (Fig S1E). Thus, the temporal profile of NMDAR-mediated protein synthesis, particularly the translation activation at 20 min, was dependent on calcium from L-VGCCs, RyRs, and SOCE (Figs 1K and S1F).

### NMDAR-mediated changes in phosphorylation of eukaryotic translation elongation factor 2 (eEF2) are differentially regulated by calcium sources

Activity-mediated protein synthesis in neurons is tightly coordinated through translation inhibition and activation pathways. To obtain mechanistic insights into these two components, we chose to investigate translation inhibition and activation using separate readouts. eEF2, an essential and obligatory factor for ribosome translation on mRNA, is phosphorylated by eEF2 kinase in a calcium-dependent manner (48, 49). The increase in the phosphorylation of eEF2 (p-eEF2) decreases translation elongation and acts as a reliable marker for global protein synthesis inhibition (Fig 2A). As reported previously (25, 30), NMDAR stimulation of primary neurons increased eEF2 phosphorylation at 1 min, after which the phospho-eEF2 levels decreased (Figs 2B and D and S2A and B). As observed in our previous studies (25, 30), at 5-min NMDAR stimulation, eEF2 phosphorylation seemed lower than 1 min and higher compared with untreated conditions, indicating a net translation inhibition (Figs 2B and D and S2A and B). The p-eEF2 data at 5-min NMDAR stimulation correlates with the FUNCAT data indicating a

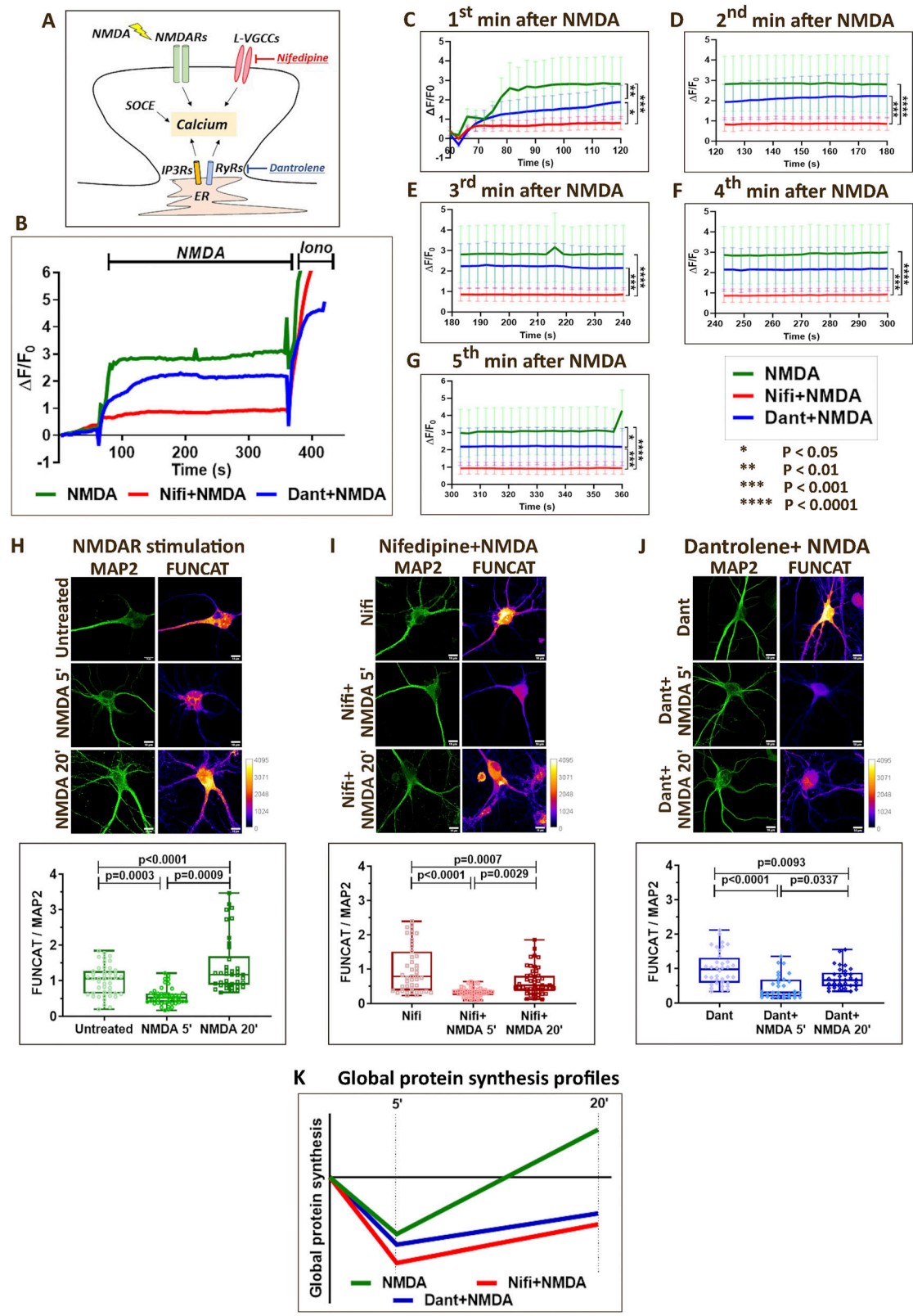

**Figure 1. NMDAR-mediated protein synthesis is regulated by calcium signals generated by distinct sources.**
**(A)** Model depicting the different sources, which contribute to the increase in cytosolic calcium on NMDAR stimulation. The respective agonists and antagonists of the calcium channels are indicated. **(B)** Calcium imaging traces from DIV15 primary cortical neurons indicating the change in Fluo-4 AM fluorescence ($\Delta F/F0$) on NMDAR stimulation (20 $\mu$M NMDA) in the presence of nifedipine (50 $\mu$M) and dantrolene (60 $\mu$M). Data are represented as the mean. N = 6–9 experiments. **(C, D, E, F, G)**

decrease in protein synthesis (Fig 1H). At 20-min NMDAR stimulation, we observed a decrease in eEF2 phosphorylation compared with the basal condition, resulting in a further reduction in translation inhibition and a net increase in protein synthesis (Figs 2B and D and S2A and B), which was also captured by the FUNCAT data (Fig 1H). In the presence of the NMDAR channel antagonist MK801, the eEF2 phosphorylation profile was completely blocked, indicating that the response was specific and initiated by calcium through NMDARs (Fig S2A). Similarly, the changes in eEF2 phosphorylation were abolished in the absence of extracellular calcium, showing that the calcium influx from an external source was the trigger for NMDAR-mediated translation response (Fig S2B).

NMDAR stimulation in the presence of nifedipine (L-VGCC antagonist) did not affect the increase in eEF2 phosphorylation at 1 min (Fig 2B). However, in the presence of nifedipine, the phospho-eEF2 levels did not decrease after 1 min and continued to persist at a higher level until 20 min (Fig 2B). Thus, blocking L-VGCCs led to increased eEF2 phosphorylation at 5 and 20 min as compared to NMDAR stimulation in the absence of nifedipine (Fig 2B). These results were further validated through immunostaining for p-eEF2. NMDAR stimulation for 5 min caused an increase in p-eEF2 levels in MAP2-positive neurons (Fig 2C), and the presence of nifedipine further increased the p-eEF2 as compared to NMDAR stimulation alone (Fig 2C). In the presence of dantrolene (RyR antagonist), NMDAR-mediated eEF2 phosphorylation was not affected at 1- and 5-min time points (Fig 2D). At 20-min time point, although NMDAR stimulation alone led to a decrease in eEF2 phosphorylation, it remained significantly higher in the presence of dantrolene (Fig 2D). Interestingly, immunostaining experiments showed that 5-min NMDAR stimulation in the presence of dantrolene caused a significant increase in p-eEF2 levels as compared to NMDAR stimulation alone (Fig 2E). The use of 2-APB (SOCE blocker) affected the NMDAR-mediated eEF2 phosphorylation mainly at 5 min, where p-eEF2 levels were higher in the presence of 2-APB (Fig S2C and D). However, eEF2 phosphorylation returned to basal levels at 20-min NMDAR stimulation in the presence of 2-APB (Fig S2C). In summary, calcium through L-VGCCs, RyRs, and SOCE are important contributors in regulating eEF2 phosphorylation at 5-min NMDAR stimulation (Figs 2F and S2E). However, L-VGCCs and RyRs seem more

important for the reduction in eEF2 phosphorylation and increased translation at 20 min of NMDAR stimulation (Figs 2F and S2E).

## NMDAR-mediated changes in ribosomal protein S6 (RPS6) phosphorylation are regulated by different calcium sources

We investigated the phosphorylation status of RPS6 as the readout for translation activation (Fig 3A). Although mTOR is the well-known upstream regulator of RPS6 phosphorylation, reports suggest that S6 kinase activity and RPS6 phosphorylation can be regulated in an Akt- and a calcium-dependent manner as well (50, 51, 52, 53, 54), thus making it an interesting readout for NMDAR-mediated translation activation. NMDAR stimulation of primary cortical neurons showed an increase in RPS6 phosphorylation primarily at 20-min time point indicating translation activation (Fig 3B and D). This was in correlation with the increased FUNCAT signal (Fig 1) and decreased eEF2 phosphorylation (Fig 2) at 20-min NMDAR stimulation. At 5-min NMDAR stimulation, p-RPS6 levels showed a robust increase in MAP2-positive neurons through immunostaining experiments (Fig 3C and E), though this was captured only as a trend of increase in Western blotting (Fig 3B and D). Thus, 5-min NMDAR stimulation correlates with active reduction in translation inhibition (indicated through eEF2 phosphorylation), as well as potential translation activation.

The pre-treatment of neurons with nifedipine (L-VGCC antagonist) prevented the NMDAR-mediated increase in RPS6 phosphorylation at the 20-min time point (Fig 3B). Immunostaining experiments indicated that nifedipine blocked the NMDAR-mediated p-RPS6 increase at 5-min time point as well (Fig 3C). Similar results were observed with pre-treatment of dantrolene (RyR antagonist) and 2-APB (SOCE blocker). Dantrolene and 2-APB blocked the NMDAR-mediated p-RPS6 increase at 20 min as indicated by immunoblotting (Figs 3D and S3A) and 5 min as indicated by immunostaining (Figs 3E and S3B). These results confirm that calcium through L-VGCCs, RyRs, and SOCE was important for increased RPS6 phosphorylation and translation activation on NMDAR stimulation (Figs 3F and S3C).

Results from earlier sections established that L-VGCCs were critical for both reducing translation inhibition (eEF2 phosphorylation) and increasing translation activation (RPS6 phosphorylation).

---

Quantification of change in Fluo-4 AM fluorescence (ΔF/F0) on NMDAR stimulation (20 µM NMDA) in the presence of nifedipine (50 µM) and dantrolene (60 µM). Data are represented as the mean ± SEM. N = 6–9 independent experiments. Two-way ANOVA followed by Tukey's multiple comparison test. **(C)** First-minute (60–120 s) quantification. $P < 0.0001$ for NMDA versus Nifi+NMDA, $P = 0.0008$ for NMDA versus Dant+NMDA, $P = 0.0428$ for Nifi+NMDA versus Dant+NMDA. **(D)** Second-minute (123–180 s) quantification. $P < 0.0001$ for NMDA versus Nifi+NMDA, $P = 0.0003$ for Nifi+NMDA versus Dant+NMDA. **(E)** Third-minute (183–240 s) quantification. $P < 0.0001$ for NMDA versus Nifi+NMDA, $P = 0.0001$ for Nifi+NMDA versus Dant+NMDA. **(F)** Fourth-minute (243–300 s) quantification. $P < 0.0001$ for NMDA versus Nifi+NMDA, $P = 0.0003$ for Nifi+NMDA versus Dant+NMDA. **(G)** Fifth-minute (300–360 s) quantification. $P < 0.0001$ for NMDA versus Nifi+NMDA, $P = 0.0139$ for NMDA versus Dant+NMDA, $P = 0.0003$ for Nifi+NMDA versus Dant+NMDA. **(H)** *Top*: Representative images of DIV15 primary cortical neurons indicating FUNCAT and MAP2 intensities on NMDAR stimulation (20 µM NMDA) for 5 and 20 min. Scale bar—10 µm. *Bottom*: Quantification of the FUNCAT intensity normalized to MAP2 intensity on NMDAR stimulation (20 µM NMDA) for 5 and 20 min. N = 38–45 neurons from four independent experiments. One-way ANOVA ($P < 0.0001$) followed by Tukey's multiple comparison test. **(I)** *Top*: Representative images of DIV15 primary cortical neurons indicating FUNCAT and MAP2 intensities on NMDAR stimulation (20 µM NMDA) for 5 and 20 min in the presence of nifedipine (50 µM). Scale bar—10 µm. *Bottom*: Quantification of the FUNCAT intensity normalized to MAP2 intensity on NMDAR stimulation (20 µM NMDA) for 5 and 20 min in the presence of nifedipine (50 µM). Each data point represents an individual neuron. N = 43–48 neurons from four independent experiments. One-way ANOVA ($P < 0.0001$) followed by Tukey's multiple comparison test. **(J)** *Top*: Representative images of DIV15 primary cortical neurons indicating FUNCAT and MAP2 intensities on NMDAR stimulation (20 µM NMDA) for 5 and 20 min in the presence of dantrolene (60 µM). Scale bar—10 µm. *Bottom*: Quantification of the FUNCAT intensity normalized to MAP2 intensity on NMDAR stimulation (20 µM NMDA) for 5 and 20 min in the presence of dantrolene (60 µM). Each data point represents an individual neuron. N = 31–36 neurons from four independent experiments. One-way ANOVA ($P < 0.0001$) followed by Tukey's multiple comparison test. **(K)** Representative graph summarizing the temporal profile of global protein synthesis on NMDAR stimulation in the presence or absence of calcium channel antagonists (nifedipine and dantrolene).

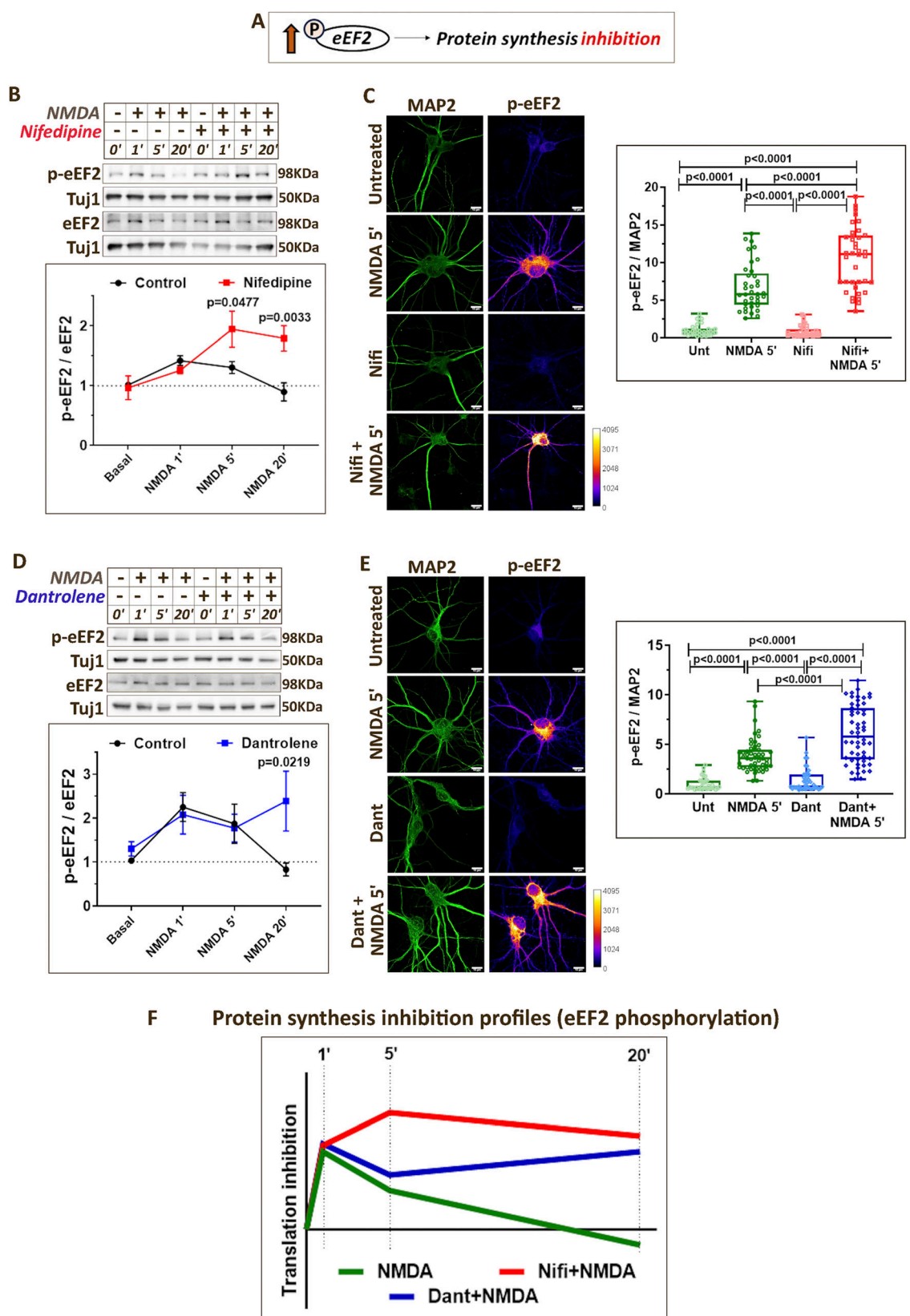

**Figure 2. NMDAR-mediated changes in eEF2 phosphorylation are differentially regulated by calcium sources.**
**(A)** Cartoon depicting the role of eEF2 phosphorylation in protein synthesis regulation. **(B)** *Top*: Representative immunoblots indicating p-eEF2, eEF2, and Tuj1 levels in DIV15 primary cortical neurons on NMDAR stimulation (20 μM NMDA) for 1, 5, and 20 min in the presence or absence of nifedipine (50 μM). *Bottom*: Quantification of the

To further validate these findings, we treated primary neurons with BayK8644, an agonist of L-VGCCs, for 5 and 20 min. We validated the activity of BayK8644 by showing that it causes calcium influx in neurons (Fig S3D). A 5-min treatment of BayK8644 resulted in an increase in eEF2 phosphorylation in neurons, which was significantly reduced by 20 min (Fig S3E). Interestingly, BayK8644 treatment also caused an increase in RPS6 phosphorylation at 20 min (Fig S3F). Thus, calcium through L-VGCCs can independently cause the translation inhibition at 5 min, which changes to global translation activation by 20 min.

### IP3-mediated calcium release and SOCE are necessary for translation activation on mGluR stimulation

To further validate the role of calcium in regulating neuronal protein synthesis, we employed an independent paradigm of mGluR stimulation. mGluRs being G protein–coupled receptors, their stimulation generates IP3, resulting in cytosolic calcium increase mediated by IP3Rs and SOCE (28, 29, 55) (Fig 4A). Stimulating the neurons with DHPG (Group 1 mGluR agonist) caused a sustained increase in cytosolic calcium for 5 min, which was completely abolished in the presence of 2-APB (IP3R and SOCE blocker) (Fig 4B and C). Activation of mGluRs is also known to cause an up-regulation of neuronal protein synthesis (25, 31, 32, 56). As reported previously, mGluR stimulation using DHPG caused an increase in FUNCAT signal at 5- and 20-min time points (Fig 4D). However, in the presence of 2-APB, mGluR stimulation failed to elicit an increase in the FUNCAT signal (Fig 4E), indicating that calcium through IP3Rs and SOCE was necessary for mGluR-mediated translation activation. We observed an increase in the MAP2 levels at 5 and 20 min of mGluR stimulation (Fig S4A), and hence did not normalize FUNCAT signals to MAP2 in these experiments. The increased translation of MAP2 on mGluR stimulation has been reported previously as well (57). The mGluR-mediated increase in MAP2 levels was also abolished in the presence of 2-APB (Fig S4B), further validating the role of calcium in mGluR mediated translation up-regulation.

Furthermore, we investigated the eEF2 and RPS6 phosphorylation responses on mGluR stimulation at 1, 5, and 20 min. DHPG caused a significant reduction in eEF2 phosphorylation at 5 and 20 min (Fig S4C) while resulting in an increase in RPS6 phosphorylation at the 5-min time point (Fig S4D). We used 2-APB to test the response at the 5-min time point through immunostaining. 5-min mGluR stimulation led to a decrease in p-eEF2 levels in MAP2-

positive neurons, which was prevented in the presence of 2-APB (Fig 4F). Similarly, mGluR-mediated increase in p-RPS6 levels was also abolished in the presence of 2-APB (Fig 4G). Hence, mGluR stimulation led to an up-regulation of protein synthesis (decrease in translation inhibition and increased activation), which was completely dependent on the cytosolic calcium increase through IP3Rs and SOCE (Fig 4H).

### Dysregulation of NMDAR-mediated calcium and protein synthesis response in Alzheimer's disease neurons

So far, our results have shown that the calcium signal has an important role in regulating protein synthesis on NMDAR and mGluR stimulation. Furthermore, we aimed to understand the relevance of this link in the context of Alzheimer's disease (AD) (58). Several studies have indicated that dysregulation of calcium homeostasis is an early phenotype in AD and a key factor in accelerating the pathological changes in AD (34, 35, 36). Defective calcium signaling involving NMDARs, L-VGCCs, RyRs, and SOCE has been implicated in AD (34, 35, 36). Considering this, we investigated the NMDAR-mediated calcium response in AD neurons and measured its effects on protein synthesis.

We used primary cortical neurons (DIV15) prepared from C57BL/6 WT and APPswe/PS1dE9 (APP/PS1) double transgenic AD mice (Fig 5A). Stimulation of NMDARs in WT mouse neurons resulted in a robust increase in cytosolic calcium (Fig 5B). However, NMDAR-mediated calcium influx was clearly reduced in APP/PS1 neurons (Fig 5B), indicating the defective calcium response in AD. Likewise, the NMDAR-mediated protein synthesis response was preserved in WT mouse neurons. WT neurons showed the biphasic translation response on NMDAR stimulation, demonstrated by the decrease in the FUNCAT signal at 5 min and an increase in the FUNCAT signal at 20 min compared with the basal condition (Fig 5C). However, the NMDAR-mediated changes in protein synthesis were completely abolished in AD, as APP/PS1 neurons showed no change in FUNCAT signal at 5 and 20 min compared with the basal condition (Fig 5C). Interestingly, the basal FUNCAT signal was significantly lower in APP/PS1 neurons compared with WT neurons (Fig 5C), displaying both basal and activity-mediated protein synthesis defects in AD.

Furthermore, we measured the NMDAR-mediated changes in eEF2 and RPS6 phosphorylation at 1, 5, and 20-min time points in the context of AD. As shown previously, the neurons from WT mice responded to NMDAR stimulation, showing an increase in eEF2 phosphorylation at 1 and 5 min, which decreased by the 20-min

p-eEF2-to-eEF2 ratio on NMDAR stimulation (20 μM NMDA) for 1, 5, and 20 min in the presence or absence of nifedipine (50 μM). Data are represented as the mean ± SEM. N = 4. Two-way ANOVA followed by Sidak's multiple comparison test. **(C)** *Left*: Representative images of DIV15 primary cortical neurons immunostained for p-eEF2 and MAP2 on 5-min NMDAR stimulation (20 μM NMDA) in the presence or absence of nifedipine (50 μM). Scale bar—10 μm. *Right*: Quantification of p-eEF2 intensity normalized to MAP2 intensity on 5-min NMDAR stimulation (20 μM NMDA) in the presence or absence of nifedipine (50 μM). Each data point represents an individual neuron. N = 37–40 neurons from three independent experiments. One-way ANOVA (P < 0.0001) followed by Tukey's multiple comparison test. **(D)** *Top*: Representative immunoblots indicating p-eEF2, eEF2, and Tuj1 levels in DIV15 primary cortical neurons on NMDAR stimulation (20 μM NMDA) for 1, 5, and 20 min in the presence or absence of dantrolene (60 μM). *Bottom*: Quantification of the p-eEF2-to-eEF2 ratio on NMDAR stimulation (20 μM NMDA) for 1, 5, and 20 min in the presence or absence of dantrolene (60 μM). Data are represented as the mean ± SEM. N = 6. Two-way ANOVA followed by Sidak's multiple comparison test. **(E)** *Left*: Representative images of DIV15 primary cortical neurons immunostained for p-eEF2 and MAP2 on 5-min NMDAR stimulation (20 μM NMDA) in the presence or absence of dantrolene (60 μM). Scale bar—10 μm. *Right*: Quantification of p-eEF2 intensity normalized to MAP2 intensity on 5-min NMDAR stimulation (20 μM NMDA) in the presence or absence of dantrolene (60 μM). Each data point represents an individual neuron. N = 34–55 neurons from three independent experiments. One-way ANOVA (P < 0.0001) followed by Tukey's multiple comparison test. **(F)** Graph summarizing the temporal profile of eEF2 phosphorylation indicating protein synthesis inhibition on NMDAR stimulation in the presence or absence of calcium channel antagonists (nifedipine and dantrolene).

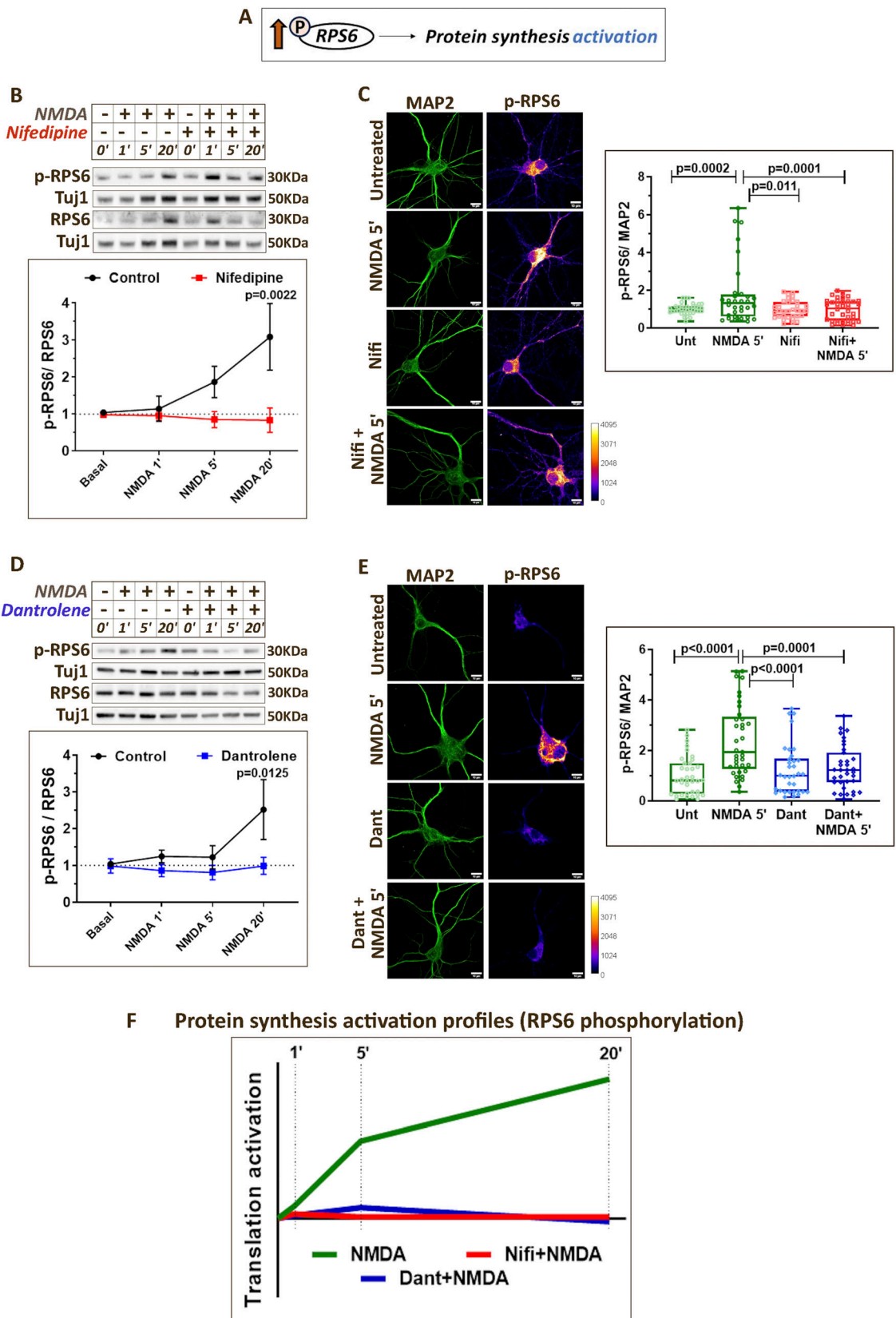

**Figure 3. NMDAR-mediated changes in RPS6 phosphorylation are regulated by different calcium sources.**
**(A)** Cartoon depicting the role of RPS6 phosphorylation in protein synthesis regulation. **(B)** *Top*: Representative immunoblots indicating p-RPS6, RPS6, and Tuj1 levels in DIV15 primary cortical neurons on NMDAR stimulation (20 µM NMDA) for 1, 5, and 20 min in the presence or absence of nifedipine (50 µM). *Bottom*: Quantification of the

time point (Fig S5A). Accordingly, the RPS6 phosphorylation increased at 20-min NMDAR stimulation in WT neurons (Fig S5B), depicting the translation activation at the 20-min time point. However, the NMDAR-mediated changes in eEF2 and RPS6 phosphorylation were completely dysregulated in AD neurons (Fig S5C and D). APP/PS1 neurons showed a decrease in eEF2 phosphorylation at 1-min NMDAR stimulation, which remained low until the 20-min time point (Fig S5C), whereas RPS6 phosphorylation showed no changes with 1, 5, and 20 min of stimulation (Fig S5D). Thus, the correlation and crosstalk between eEF2 phosphorylation, RPS6 phosphorylation, and protein synthesis (FUNCAT) were perturbed in the AD neurons, indicating an overall dysregulation of basal and NMDAR-mediated translation response (Fig S5E).

Finally, we validated our findings in a human stem cell–derived neuron model of AD. We used iPSCs generated from familial AD patients carrying PSEN1 L150P mutation and its corresponding gene-corrected isogenic control line PSEN1 L150P GC (Fig 5D). The mutant and control iPSCs were characterized for the expression of stem cell markers OCT4, NANOG, SSEA4, and TRA-160 (Fig S5F and G). The iPSCs were differentiated into forebrain glutamatergic neurons (Fig 5D). The intermediate neural stem cell (NSC) stage was characterized using Nestin and Pax6 (Fig S5H and I), whereas the neurons were characterized for dendritic marker MAP2 and axonal marker Tau (Fig S5J and K). The differentiated neurons were subjected to calcium imaging on NMDAR stimulation. Control (PSEN1 L150P GC) neurons showed a robust increase in cytosolic calcium on NMDAR stimulation (Fig 5E), whereas the mutant (PSEN1 L150P) neurons completely lacked the calcium response on the addition of NMDA (Fig 5E). Accordingly, the NMDAR-mediated changes in protein synthesis were also affected in AD neurons. Although the control (PSEN1 L150P GC) neurons showed an increase in eEF2 phosphorylation at 5-min NMDAR stimulation (Fig 5F), the mutant (PSEN1 L150P) neurons showed no changes in eEF2 phosphorylation (Fig 5G). Thus, our results show that calcium is an important regulator of protein synthesis on neuronal activity. AD neurons show perturbation of both calcium and translation response on NMDAR stimulation.

## Discussion

Our study establishes a clear link between calcium signals and activity-mediated protein synthesis in neurons. We have focused on two classes of glutamate receptors that play an important role in synaptic plasticity, namely, NMDARs and mGluRs. Activation of

these receptors generates distinct calcium profiles involving multiple calcium channels on the plasma membrane and ER (2, 3, 5). To probe the link between calcium and activity-mediated protein synthesis, we subjected primary cortical neurons to NMDAR and mGluR stimulation in the presence of different calcium channel blockers and measured protein synthesis using three independent readouts, namely, FUNCAT (global readout of de novo protein synthesis), eEF2 phosphorylation (marker for translation inhibition), and RPS6 phosphorylation (readout for translation activation).

Calcium response on NMDAR stimulation involves NMDARs itself as the first source, sustained by L-VGCCs and CICR from the ER predominantly mediated by RyRs (5, 26). Accordingly, we observe that nifedipine (L-VGCC antagonist) and dantrolene (RyR antagonist) significantly reduce the NMDAR-mediated calcium increase in the cytosol (Fig 1). Although nifedipine reduced the NMDAR-mediated calcium for the entire 5-min duration that we monitored, dantrolene significantly affected the calcium response at the first- and fifth-minute time points (Fig 1). Thus, starting from the first minute, NMDARs, L-VGCCs, and RyRs contribute to the cytosolic calcium increase on NMDAR stimulation.

Previous work from our laboratory has shown that NMDAR stimulation in neurons generates a biphasic protein synthesis response involving a shift from translation inhibition to the activation phase (30). Here, on measuring the protein synthesis readouts across 20 min, we have further resolved the translation response into three components, which constitute the final biphasic response—increased translation inhibition (first minute), decrease in translation inhibition (1–5 min), and increased translation activation (5–20 min) (Fig 6A). 1-min stimulation of NMDARs results in a robust increase in eEF2 phosphorylation indicating the phase of increased translation inhibition. Correspondingly, the FUNCAT signal is decreased at 1-min NMDAR stimulation showing the down-regulation of global protein synthesis (Fig 6A). At 5-min NMDAR stimulation, RPS6 phosphorylation begins to increase, and eEF2 phosphorylation is lower than the 1-min time point, though it remains higher than the basal condition. Thus, the duration from 1 to 5 min corresponds to the phase of decrease in translation inhibition (Fig 6A). However, the FUNCAT signal is decreased at 5-min NMDAR stimulation compared with the basal condition, demonstrating that the net global protein synthesis is still low in the neurons (Fig 6A). The 20-min time point of NMDAR stimulation shows a robust increase in RPS6 phosphorylation and a decrease in eEF2 phosphorylation, corresponding to the phase of increased translation activation. In agreement, the FUNCAT signal increases at

---

p-RPS6-to-RPS6 ratio on NMDAR stimulation (20 μM NMDA) for 1, 5, and 20 min in the presence or absence of nifedipine (50 μM). Data are represented as the mean ± SEM. N = 4. Two-way ANOVA followed by Sidak's multiple comparison test. **(C)** *Left*: Representative images of DIV15 primary cortical neurons immunostained for p-RPS6 and MAP2 on 5-min NMDAR stimulation (20 μM NMDA) in the presence or absence of nifedipine (50 μM). Scale bar—10 μm. *Right*: Quantification of p-RPS6 intensity normalized to MAP2 intensity on 5-min NMDAR stimulation (20 μM NMDA) in the presence or absence of nifedipine (50 μM). Each data point represents an individual neuron. N = 32–38 neurons from three independent experiments. One-way ANOVA (*P* = 0.0003) followed by Tukey's multiple comparison test. **(D)** *Top*: Representative immunoblots indicating p-RPS6, RPS6, and Tuj1 levels in DIV15 primary cortical neurons on NMDAR stimulation (20 μM NMDA) for 1, 5, and 20 min in the presence or absence of dantrolene (60 μM). *Bottom*: Quantification of the p-RPS6-to-RPS6 ratio on NMDAR stimulation (20 μM NMDA) for 1, 5, and 20 min in the presence or absence of dantrolene (60 μM). Data are represented as the mean ± SEM. N = 6. Two-way ANOVA followed by Sidak's multiple comparison test. **(E)** *Left*: Representative images of DIV15 primary cortical neurons immunostained for p-RPS6 and MAP2 on 5-min NMDAR stimulation (20 μM NMDA) in the presence or absence of dantrolene (60 μM). Scale bar—10 μm. *Right*: Quantification of p-RPS6 intensity normalized to MAP2 intensity on 5-min NMDAR stimulation (20 μM NMDA) in the presence or absence of dantrolene (60 μM). Each data point represents an individual neuron. N = 34–37 neurons from three independent experiments. One-way ANOVA (*P* = 0.0003) followed by Tukey's multiple comparison test. **(F)** Graph summarizing the temporal profile of RPS6 phosphorylation indicating protein synthesis activation on NMDAR stimulation in the presence or absence of calcium channel antagonists (nifedipine and dantrolene).

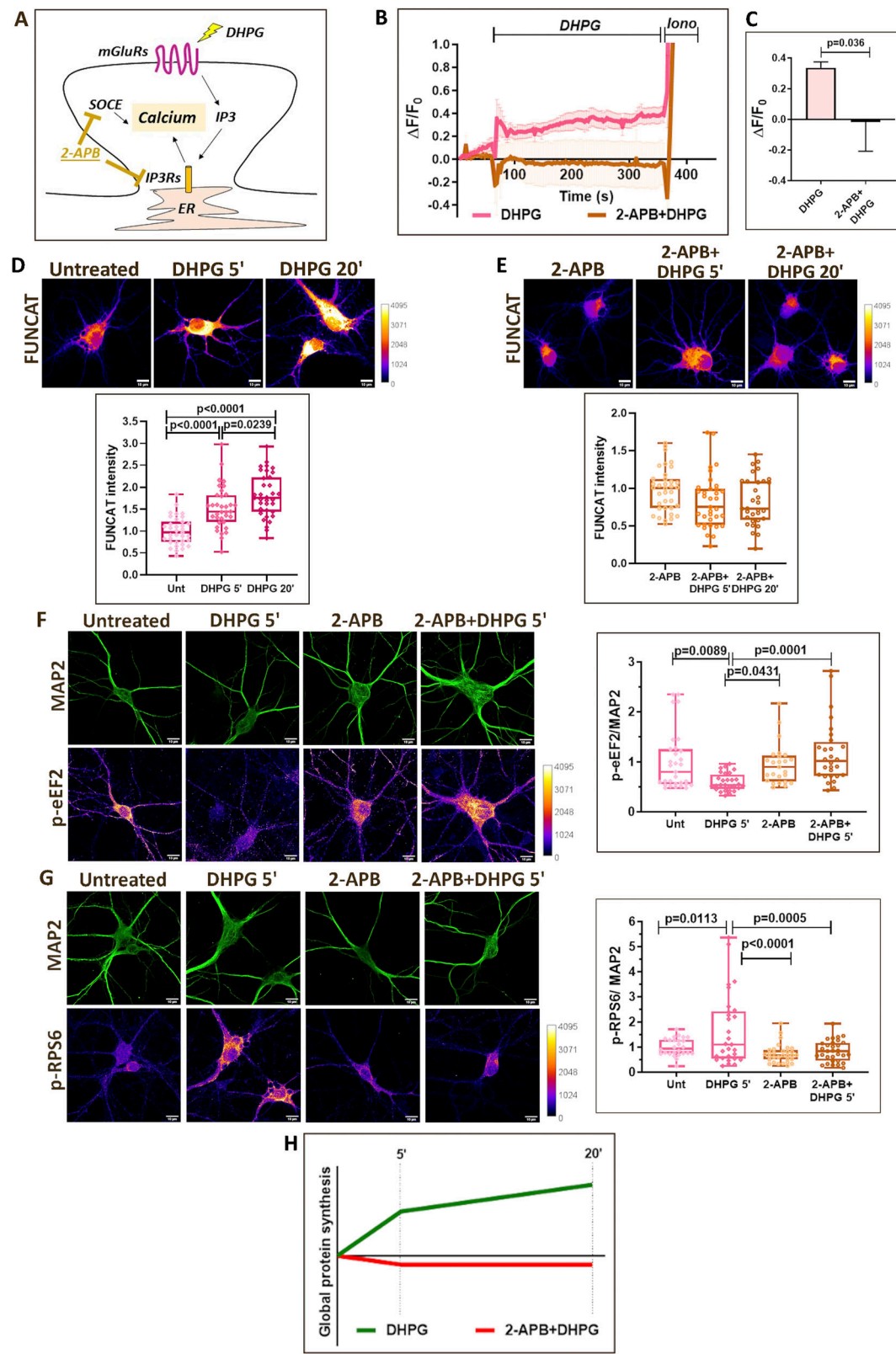

**Figure 4. IP3-mediated calcium release and SOCE are necessary for translation activation on mGluR stimulation.**
**(A)** Cartoon depicting IP3-mediated calcium release and SOCE on mGluR stimulation. **(B)** Calcium imaging traces from DIV15 primary cortical neurons indicating the change in Fluo-4 AM fluorescence (ΔF/F0) on mGluR stimulation (50 μM DHPG) in the presence or absence of 2-APB (50 μM). Data are represented as the mean ± SEM. N = 3–5. **(C)** Quantification of the change in Fluo-4 AM fluorescence (ΔF/F0) for the duration of 5 min after the addition of DHPG (50 μM) in the presence or absence of 2-APB

20-min NMDAR stimulation compared with the basal condition, implying up-regulation of global protein synthesis (Fig 6A).

We show that calcium influx through NMDARs is necessary for generating the overall biphasic translation profile as the presence of MK801 (NMDAR antagonist) or the absence of extracellular calcium completely abolishes the NMDAR-mediated translation response. The increased eEF2 phosphorylation at 1-min NMDAR stimulation is affected only in the presence of MK801, thus making the increased translation inhibition phase solely dependent on calcium through NMDARs (Fig 6A). Calcium from L-VGCCs is important for the other two phases—reducing the translation inhibition (eEF2 phosphorylation) at 5 min and promoting the translation activation (RPS6 phosphorylation) at 20 min (Fig 6A). This is based on our observation that in the presence of nifedipine (L-VGCC antagonist), eEF2 phosphorylation does not reduce at 5 and 20 min of NMDAR stimulation, and p-RPS6 levels do not increase at the 20-min time point. Accordingly, the FUNCAT signal remains low at 20-min NMDAR stimulation in the presence of nifedipine. We also capture this function of L-VGCCs using its agonist BayK8644. Treatment of neurons with BayK8644 increases eEF2 phosphorylation at 5-min time point, which decreases by 20 min. Correspondingly, BayK8644 treatment increases RPS6 phosphorylation at 20-min time point, similar to NMDAR stimulation. Calcium from RyRs predominantly influences the NMDAR-mediated translation activation at 20 min as p-eEF2 levels remain high, p-RPS6 does not increase, and the FUNCAT signal remains low in the presence of dantrolene (RyR antagonist) (Fig 6A). Because L-VGCCs affected the reduction in translation inhibition more starkly than RyRs, we hypothesize that calcium through L-VGCCs acts as an upstream regulator of NMDAR-mediated protein synthesis than RyRs. However, the complex interplay between these two channels in the regulation of NMDAR-mediated translation activation needs to be further explored.

The other important routes in neuronal calcium signaling include IP3Rs on the ER and SOCE. SOCE is triggered on ER store depletion and involves calcium influx into the cytosol through various channels on the plasma membrane, predominantly Orai, to refill the ER (59). Although few studies have reported the involvement of SOCE in NMDAR-mediated calcium response (26, 46, 60, 61), the role of IP3Rs in the context of NMDAR stimulation is largely unexplored. In our study, we used 2-APB, which is reported to block IP3Rs and SOCE (62, 63, 64) to investigate the role of both these pathways in NMDAR-mediated translation response. In the

presence of 2-APB, NMDAR stimulation resulted in hyper-phosphorylation of eEF2 at 5 min which recovered by 20 min. However, NMDAR-mediated translation activation and increase in RPS6 phosphorylation at 20 min were significantly perturbed in the presence of 2-APB. Thus, ER-mediated calcium release from RyRs and IP3Rs, as well as ER calcium replenishment through SOCE, is important for the translation up-regulation phase of NMDAR stimulation (Fig 6A).

It is of interest to note that the initial inhibition of NMDAR-mediated protein synthesis is primarily controlled by inhibition at the translation elongation step (increased eEF2 phosphorylation and no change in RPS6 phosphorylation). However, the later phase of protein synthesis activation involves regulation at both translation initiation and elongation steps (decreased eEF2 phosphorylation and increased RPS6 phosphorylation). Overall, the initial inhibition of global translation along with protein degradation (25, 30), followed by robust protein synthesis activation, might be important for changing the translatome on NMDAR stimulation. These global changes in the proteome are likely to be essential for NMDAR-mediated changes in synaptic plasticity (65) and require further detailed investigation. Calcium is a well-known player in NMDAR-mediated plasticity, both in long-term potentiation and in long-term depression (66, 67, 68). Considering the polarity of neurons and their complex spatio-temporal regulation of protein synthesis and synaptic plasticity, the different calcium channels and their corresponding calcium signaling become ideal candidates to bring about these regulations. Especially in the case of calcium-permeable channels like NMDARs, calcium could be the primary player in changing the translatome and linking this to plasticity.

To further explore the link between calcium signal and activity-mediated protein synthesis in neurons, we used another independent paradigm of mGluR stimulation. mGluRs, belonging to the GPCR family of receptors, are known to generate IP3 and cause cytosolic calcium increase through IP3Rs and SOCE (28, 29). IP3-mediated calcium release and SOCE are also shown to be essential for mGluR-mediated synaptic plasticity (29). The protein synthesis response on mGluR stimulation is also studied extensively, where mGluRs are reported to activate neuronal protein synthesis, primarily mediated through the mTOR pathway (25, 31, 56). Connecting these two aspects, we investigated the role of calcium through IP3Rs and SOCE in mGluR-mediated translation response. We observe that 2-APB (IP3R and SOCE blocker)

(50 µM). Data are represented as the mean ± SEM. N = 3–5. Unpaired t test. **(D)** *Top*: Representative images of DIV15 primary cortical neurons indicating the FUNCAT intensity on mGluR stimulation (50 µM DHPG) for 5 and 20 min. Scale bar—10 µm. *Bottom*: Quantification of the FUNCAT intensity on mGluR stimulation (50 µM DHPG) for 5 and 20 min. Each data point represents an individual neuron. N = 35 neurons from three independent experiments. One-way ANOVA (P < 0.0001) followed by Tukey's multiple comparison test. **(E)** *Top*: Representative images of DIV15 primary cortical neurons indicating the FUNCAT intensity on mGluR stimulation (50 µM DHPG) for 5 and 20 min in the presence of 2-APB (50 µM). Scale bar—10 µm. *Bottom*: Quantification of the FUNCAT intensity on mGluR stimulation (50 µM DHPG) for 5 min and 20 min in the presence of 2-APB (50 µM). Each data point represents an individual neuron. N = 32–37 neurons from three independent experiments. One-way ANOVA followed by Tukey's multiple comparison test. **(F)** *Left*: Representative images of DIV15 primary cortical neurons immunostained for p-eEF2 and MAP2 on 5-min mGluR stimulation (50 µM DHPG) in the presence or absence of 2-APB (50 µM). Scale bar—10 µm. *Right*: Quantification of p-eEF2 intensity normalized to MAP2 intensity on 5-min mGluR stimulation (50 µM DHPG) in the presence or absence of 2-APB (50 µM). Each data point represents an individual neuron. N = 25–31 neurons from two independent experiments. One-way ANOVA (P = 0.0002) followed by Tukey's multiple comparison test. **(G)** *Left*: Representative images of DIV15 primary cortical neurons immunostained for p-RPS6 and MAP2 on 5-min mGluR stimulation (50 µM DHPG) in the presence or absence of 2-APB (50 µM). Scale bar—10 µm. *Right*: Quantification of p-RPS6 intensity normalized to MAP2 intensity on 5-min mGluR stimulation (50 µM DHPG) in the presence or absence of 2-APB (50 µM). Each data point represents an individual neuron. N = 26–31 neurons from two independent experiments. One-way ANOVA (P < 0.0001) followed by Tukey's multiple comparison test. **(H)** Representative graph summarizing the temporal profile of global protein synthesis on mGluR stimulation in the presence or absence of calcium channel antagonist 2-APB.

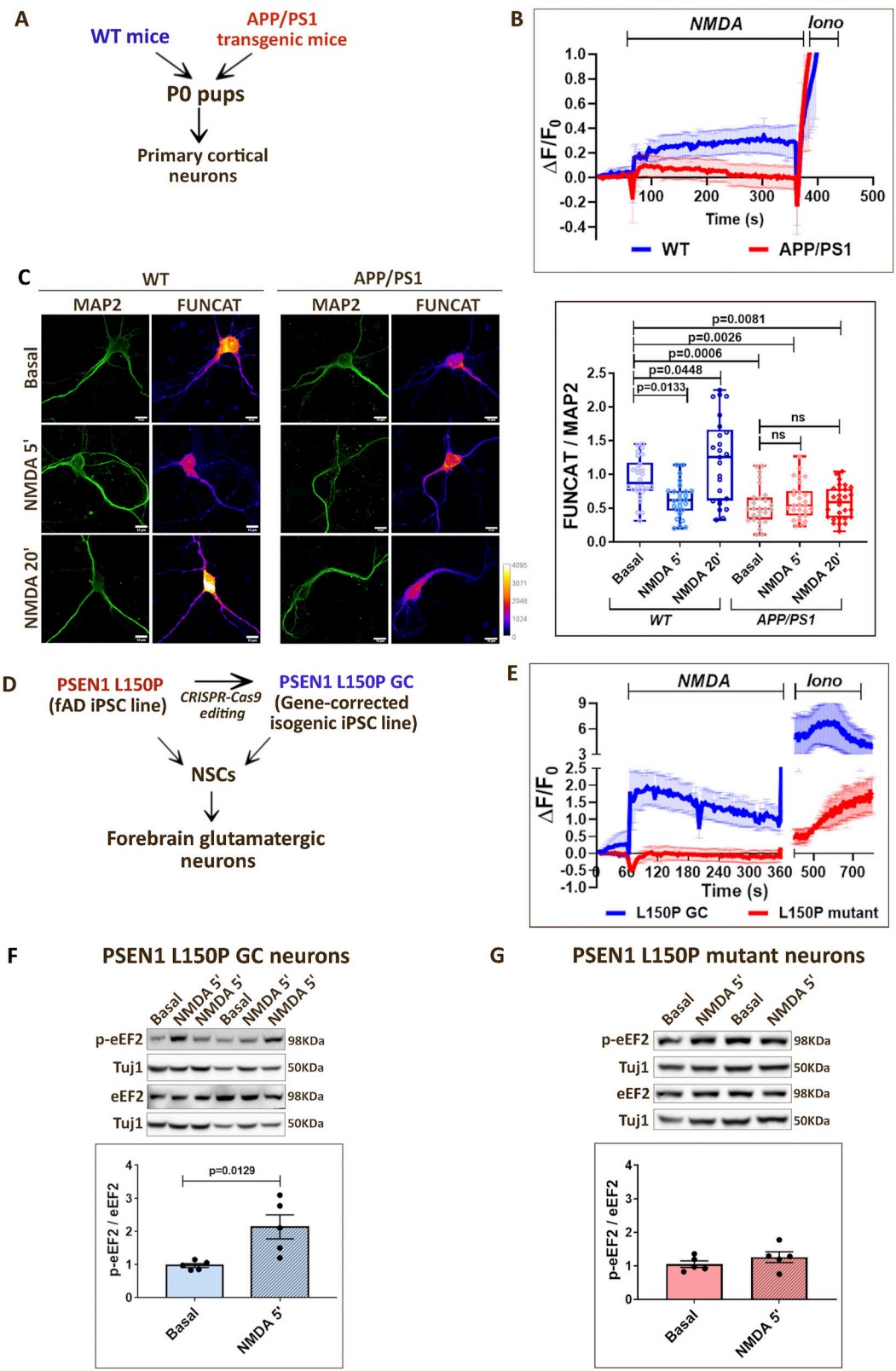

**Figure 5. Dysregulation of NMDAR-mediated calcium and protein synthesis response in Alzheimer's disease neurons.**
**(A)** Model depicting the WT and AD mouse models used in the study. **(B)** Calcium imaging traces from WT and APP/PS1 mouse primary neurons indicating the change in Fluo-4 AM fluorescence ($\Delta F/F0$) on NMDAR stimulation (20 $\mu$M NMDA). Data are represented as the mean ± SEM. N = 4. **(C)** *Left*: Representative images of DIV15 primary cortical neurons from WT and APP/PS1 mice indicating FUNCAT and MAP2 intensities on NMDAR stimulation (20 $\mu$M NMDA) for 5 and 20 min. Scale bar—10 $\mu$m. *Right*:

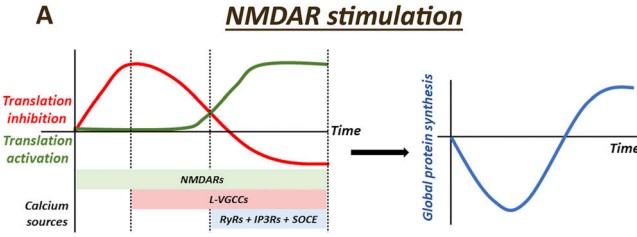

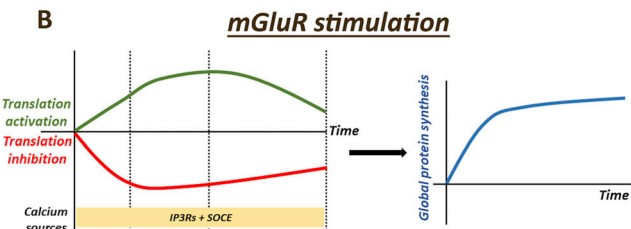

**Figure 6. Model depicting the link between calcium and activity-mediated protein synthesis in neurons.**
**(A)** Model depicting the link between calcium and translation response on NMDAR stimulation in neurons. NMDAR stimulation generates a biphasic translation response broadly involving three phases—increased translation inhibition, reduction in translation inhibition, and increased translation activation. Calcium through NMDARs is important for all three phases of the translation response, and also the sole contributor to the increase in translation inhibition. Calcium through L-VGCCs acts as a switch, essential for reducing the translation inhibition and promoting translation activation. Calcium release from ER (RyRs and IP3Rs) and SOCE is predominantly required for the increased translation activation phase. **(B)** Model depicting the link between calcium and translation response on mGluR stimulation in neurons. Stimulation of mGluRs generates a translation response involving a sustained reduction in translation inhibition and an increase in translation activation. Calcium through IP3Rs and SOCE is necessary and essential for generating this response.

completely diminishes the mGluR-mediated cytosolic calcium increase in the neurons. Global protein synthesis, as measured by the FUNCAT signal, increases with 5 and 20 min of mGluR stimulation (Fig 6B). This is reflected in eEF2 and RPS6 phosphorylation profiles, where p-eEF2 levels are reduced at 5 and 20 min, and p-RPS6 significantly increases at 5 min (Fig 6B). The presence of 2-APB completely blocked the mGluR-mediated translation activation as reflected in FUNCAT, eEF2, and RPS6 phosphorylation (Fig 6B). We observe that translation activation on mGluR stimulation is very robust as MAP2, a normalizing control used for FUNCAT experiments, also showed an increase on mGluR stimulation. Interestingly, the increase in MAP2 levels was also blocked in the presence of 2-APB. Thus, the calcium release from IP3Rs and SOCE is critical for the activation of protein synthesis on mGluR stimulation (Fig 6B).

We notice certain differences in the data obtained from Western blotting and immunostaining experiments. We observe an increase in RPS6 phosphorylation at 5-min NMDAR stimulation through immunostaining experiments, whereas we do not capture this through Western blotting. Similarly, we observed that dantrolene affects NMDAR-mediated eEF2 phosphorylation at the 5-min time point through immunostaining, but not through Western blotting. This could be due to technical differences in the way the assays work; Western blotting is processed through batch treatment and lysis of all the neurons in the plate, whereas we select MAP2-positive neurons in case of immunostaining experiments. In addition, Tuj1 is used as the normalizing factor for Western blotting, whereas we used MAP2 in imaging experiments. Nonetheless, these differences do not affect the overall interpretation of results and conclusions from the study.

Finally, we highlight the importance of the calcium translation link by showing its dysregulation in the context of AD. AD, the most common form of adult dementia, is a progressive neurodegenerative disorder leading to loss of memory and cognition (58). Synaptic dysfunction is considered to be an early phenotype and the best correlate for cognitive loss in AD (69, 70). Though Aβ plaques and neurofibrillary tangles are the hallmarks of AD, defects in many cellular processes such as mitochondrial dysfunction, and oxidative stress are identified as early contributors to AD pathology (58). Among these, disruption of calcium homeostasis is also recognized as an important early phenotype in AD. AD neurons are shown to have defective calcium signaling involving multiple channels and sensors like NMDARs, L-VGCCs, STIM1, RyRs, and SOCE (33, 34, 36, 71). Specific blockers against these calcium channels are shown to protect the neurons from Aβ toxicity as well (71, 72). Using neurons from transgenic AD mice (APP/PS1 mice) and human stem cell models of fAD (PSEN1 L150P mutant), we highlight that AD neurons have significantly reduced calcium influx on NMDAR stimulation (Fig 5). Accordingly, NMDAR-mediated protein synthesis response is also defective in AD neurons (Fig 5). Interestingly, we observe that basal protein synthesis itself is low in neurons from the APP/PS1 AD mouse model (Fig 5). Previously, many studies have reported impaired protein synthesis in AD, including the dysregulation of translation initiation and elongation factors (44, 73, 74, 75, 76); yet, the defect in activity-mediated translation is not investigated in AD. In this context, our results demonstrating the NMDAR-mediated calcium and translation defect in AD neurons would help in explaining the dysregulated synaptic signaling and plasticity in AD. Hence, we present a new perspective that calcium from distinct sources can differentially regulate the activity-mediated protein synthesis response in neurons. This mechanism is likely to play an important role in synaptic plasticity, and its dysregulation is an important contributor to the synaptic pathology in AD.

Quantification of the FUNCAT intensity normalized to MAP2 intensity on NMDAR stimulation (20 µM NMDA) for 5 and 20 min in WT and APP/PS1 mouse neurons. Each data point represents an individual neuron. N = 25–27 neurons from two independent experiments. One-way ANOVA (P < 0.0001) followed by Sidak's multiple comparison test. **(D)** Model depicting the iPSC lines used in the study and the protocol for neuronal differentiation. **(E)** Calcium imaging traces from PSEN1 L150P GC and L150P mutant neurons indicating the change in Fluo-4 AM fluorescence (ΔF/F0) on NMDAR stimulation (20 µM NMDA). Data are represented as the mean ± SEM. N = 3. **(F)** *Top*: Representative immunoblots indicating p-eEF2, eEF2, and Tuj1 levels in PSEN1 L150P GC neurons on 5-min NMDAR stimulation (20 µM NMDA). *Bottom*: Quantification of the p-eEF2-to-eEF2 ratio in PSEN1 L150P GC neurons on 5-min NMDAR stimulation (20 µM NMDA). Data are represented as the mean ± SEM. N = 5. Unpaired t test. **(G)** *Top*: Representative immunoblots indicating p-eEF2, eEF2, and Tuj1 levels in PSEN1 L150P mutant neurons on 5-min NMDAR stimulation (20 µM NMDA). *Bottom*: Quantification of the p-eEF2-to-eEF2 ratio in PSEN1 L150P mutant neurons on 5-min NMDAR stimulation (20 µM NMDA). Data are represented as the mean ± SEM. N = 5. Unpaired t test.

# Materials and Methods

## Ethics statement

All animal work was carried out following the Institutional Guidelines for the Care and Use of Laboratory Animals under the approval of the Institutional Animal Ethics Committee and the Institutional Biosafety Committee (IBSC), Centre for Brain Research, Indian Institute of Science Campus, Bangalore, India. The rats used in the study were Sprague Dawley (SD) rats. The mouse work was done with C57BL/6J mice. APPswe/PS1dE9 double transgenic mice were procured from the Jackson Laboratory (stock #34832; MMRRC) (77). APPswe/PSEN1dE9 mice carry two transgenes with AD-linked mutations: a chimeric mouse/human APP with the Swedish mutation (K670N, M671L; deletion–insertion mutation) and human PSEN1 lacking exon 9 (dE9), both under the control of the mouse prion protein promoter. The animals were housed and maintained under pathogen-free conditions in a temperature-controlled room on a 12-h light/12-h dark cycle with ad-libitum access to food and water.

All the human stem cell work was carried out as per approval from the Institutional Human Ethics Committee and Institutional Biosafety Committee at the Centre for Brain Research, Indian Institute of Science Campus, Bangalore, India.

## Rat primary neuronal cultures

Primary neuronal cultures were prepared from cerebral cortices of Sprague Dawley rat embryos (E18.5) as previously published by our laboratory (25, 30, 78). Briefly, the cortices were trypsinized for 5 min at 37°C using 0.25% trypsin (15050057; Thermo Fisher Scientific). The homogenized and dissociated cells were plated on cell culture dishes, which were pre-coated with poly-L-lysine (P2636; Sigma-Aldrich) solution (0.1 mg/ml) made in borate buffer (pH 8.5) for 5–6 h. Cells were plated at a density of 40,000–50,000 cells/cm$^2$ for biochemistry experiments and 30,000–40,000 cells/cm$^2$ for imaging-based experiments. Nitric acid–treated coverslips were used for imaging experiments. The neurons were initially plated on MEM (10095080; Thermo Fisher Scientific) supplemented with 10% FBS to aid their attachment. After 3 h, MEM was changed to Neurobasal medium (21103049; Thermo Fisher Scientific) supplemented with B27 (17504044; Thermo Fisher Scientific) and 1X GlutaMAX. The neurons were maintained in culture for 15–17 d at 37°C, with 5% $CO_2$. The neurons were supplemented with Neurobasal media every 5–6 d.

## Genotyping of mouse pups

Genotyping was performed to identify the WT and APPswe/PS1dE9 double transgenic pups (Tg APP/PS1) from the litter. A small piece of the tail was collected from the day-zero pups (P0 pups) in a sterile 1.5-ml tube and 650 $\mu$l of STE buffer (sodium chloride–Tris–EDTA buffer) (100 mM Tris, 5 mM EDTA, 200 mM NaCl, 0.2% SDS), and 20 $\mu$l of proteinase K (20 mg/ml stock solution) was added to it. The sample was incubated at 55°C for approximately 3 h with periodic vortexing to aid proper digestion. After this, the sample was vortexed for a minute and centrifuged at 14,000$g$ for 10 min

in a microcentrifuge. The supernatant was collected, and 600 $\mu$l of isopropanol was added to it. The solution was mixed, incubated on ice for 5 min, and centrifuged at 14,000$g$ for 10 min. 600 $\mu$l of 70% ethanol was added to the pellet and centrifuged at 14,000$g$ for 10 min. The DNA pellet obtained was dried at 55°C for 15 min, resuspended in 50 $\mu$l autoclaved sterile water, and incubated at 55°C for a few minutes to ensure complete dissolution. The concentration of isolated DNA was determined using a NanoDrop. Genomic DNA (50 ng) was subjected to PCR using Prp Forward Primer (0.6 $\mu$M) (5′-CCTCTTTGTGAC-TATGTGGACTGATGTCGG-3′), Prp Reverse Primer (0.8 $\mu$M) (5′-GTGGATAACCCCTCCCCCAGCCTAGACC-3′), and APP Forward Primer (0.6 $\mu$M) (5′-CCGAGATCTCTGAAGTGAAGATGGATG-3′). The PCR conditions were as follows: initial denaturation at 95°C for 1 min (1×), denaturation at 98°C for 10 s (35×), annealing at 65°C for 1 min 30 s (35×), extension at 72°C for 1 min 30 s (35×), and final extension at 72°C for 5 min (1×). The amplified PCR products were resolved on 1% agarose gel (made and run in 1× TAE buffer) through gel electrophoresis and imaged using Bio-Rad Chemi-imager.

## Mouse primary neuronal cultures

Primary neuronal cultures were prepared from cerebral cortices of day-zero pups (P0 pups) from WT (C57BL/6) and transgenic APP/PS1 (Tg APP/PS1) mice. After genotyping, the cortices from the WT and Tg pups were trypsinized for 5 min using 0.25% trypsin–EDTA at RT. The homogenized and dissociated cells were plated on dishes with pre-coated poly-L-lysine (P2636; Sigma-Aldrich). The protocol for coating and density for plating were the same as rat neurons. The neurons were plated in Neurobasal media (21103049; Thermo Fisher Scientific) supplemented with B27 (17504044; Thermo Fisher Scientific) and 1× GlutaMAX. After 4 h, the media was replaced with fresh Neurobasal media. The neurons were maintained in culture for 15–17 d at 37°C with 5% $CO_2.$ The neurons were supplemented with Neurobasal media every 5–6 d.

## iPSC maintenance and neuronal differentiation

The iPSCs were generated from the fibroblasts of a familial AD patient containing the PSEN1 L150P mutation (L150P mutant) (79). The L150P mutant iPSCs were subjected to CRISPR/Cas9-based gene editing to correct the mutation and obtain the isogenic L150P gene-corrected lines (L150P GC) (80). The iPSCs were maintained on hESC-qualified Matrigel (354277; Corning) using mTeSR1 complete media (72232; StemCell Technologies) at 37°C, 5% $CO_2$ conditions. A mixture of 1 mg/ml collagenase IV (17104019; Thermo Fisher Scientific), 0.25% trypsin, 20% knock-out serum (10828028; Thermo Fisher Scientific), and 1 mM calcium chloride (made in PBS) was used to dissociate the iPSCs for passaging.

The protocol for neural differentiation was adapted from Yichen Shi et al (81) and Yu Zhang et al (82) to differentiate iPSCs into forebrain glutamatergic neurons. The Neural Basic Media (NBM) for differentiation contained 50% DMEM/F-12 (21331–020; Thermo Fisher Scientific), 50% Neurobasal medium, 0.1% Pen-Strep, GlutaMAX, N2 (17502–048; Thermo Fisher Scientific), and

**Table 1. Drugs used.**

| Drug | Concentration (μM) | Catalog number, company |
|------|--------------------|-----------------------|
| Nifedipine | 50 | N7634; Sigma-Aldrich |
| Dantrolene | 60 | 251680; Sigma-Aldrich |
| 2-APB | 50 | 1224; Tocris |
| MK801 | 50 | 0924; Tocris |
| BayK8644 | 100 | B112; Sigma-Aldrich |
| NMDA | 20 | 0114; Tocris |
| S-DHPG | 50 | D3689; Sigma-Aldrich |
| Ionomycin | 10 | 407950; Sigma-Aldrich |

B27 without vitamin A (12587–010; Thermo Fisher Scientific). Once the iPSCs reached 70–80% confluency, they were subjected to monolayer neural induction by changing the mTeSR1 media to Neural Induction Media (NIM). The NIM is composed of NBM supplemented with small molecules SB431542 (10 μM, an inhibitor of the TGFβ pathway) (72232; StemCell Technologies) and LDN193189 (0.1 μM, an inhibitor of the BMP pathway) (72142; StemCell Technologies). The cells were subjected to neural induction for 12–15 d by changing the media every day until a uniform neuroepithelial layer had formed. After the induction, the monolayer was dissociated using Accutase (A6964; Sigma-Aldrich) and centrifuged at 1,000g for 3 min at RT. The cells were plated overnight in NIM containing 10 μM ROCK inhibitor (Y0503; Sigma-Aldrich) on pre-coated poly-L-ornithine/laminin dishes. Poly-L-ornithine (1:10 dilution in 1X PBS) (P4957; Sigma-Aldrich) coating was performed at 37°C for a minimum of 4 h, and washed thrice with 1X PBS, followed by overnight coating with laminin (5 μg/ml diluted in 1X PBS) (L2020; Sigma-Aldrich) at 37°C. The NSCs were maintained in Neural Expansion Media composed of NBM supplemented with FGF (10 ng/ml) (100-18C; Peprotech) and EGF (10 ng/ml) (AF-100-15; Peprotech). Neuronal maturation and terminal differentiation were achieved by plating the NSCs at a density of 25,000–35,000 cells/cm$^2$ in the Neural Maturation Media composed of NBM supplemented with BDNF (20 ng/ml) (450-02; Peprotech), GDNF (10 ng/ml) (450-10; Peprotech), L-ascorbic acid (200 μM) (A4403; Sigma-Aldrich), and db-Camp (50 μM) (D0627; Sigma-Aldrich). The neurons were subjected to maturation for 4–5 wk by supplementing them with Neural Maturation Media every 4–5 d.

### FUNCAT (fluorescent non-canonical amino acid tagging)

For metabolic labeling, DIV15 neurons were incubated in methionine-free DMEM (21013024; Thermo Fisher Scientific) for 30 min. After this, the neurons were treated with L-azidohomoalanine (AHA, 1 μM) (1066100; Click Chemistry Tools) for 30 min in Met-free DMEM (21013024; Thermo Fisher Scientific). During the last 10 min of the AHA treatment, the drugs nifedipine (50 μM), dantrolene (60 μM), or 2-APB (50 μM) were added (refer to Table 1 for details of the drugs). After the 30 min of AHA treatment, NMDA (20 μM) or DHPG (50 μM) was added for 5 or 20 min to stimulate NMDARs and mGluRs, respectively. After the stimulation, the

coverslips were washed with 1X PBS and fixed with 4% PFA for 15 min. After three washes with 1X PBS, the neurons were permeabilized for 10 min with 0.3% Triton X-100 solution prepared in TBS$_{50}$ (50 mM Tris, 150 mM NaCl, pH 7.6), followed by blocking for 1 h with a mixture of 2% BSA and 2% FBS prepared in TBS$_{50}$T (TBS50 with 0.1% Triton X-100). After blocking, the neurons were subjected to the FUNCAT reaction for 2 h as per the kit instructions where the AHA-incorporated proteins were tagged with an alkyne-fluorophore Alexa Fluor 555 through click reaction (C10269, CLICK-iT Cell Reaction Buffer Kit; Click Chemistry Tools). After three washes with TBS$_{50}$t, the neurons were incubated overnight with MAP2 antibody at 4°C. This was followed by secondary antibody incubation for 1 h at RT to visualize MAP2 (refer to Table 2 for details of the antibody). The coverslips were mounted using Mowiol and imaged on an Olympus FV3000 confocal laser scanning upright microscope with a 60X objective. The pinhole was kept at 1 Airy Unit and the optical zoom at 2X to satisfy Nyquist's sampling criteria in the XY-direction. The objective was moved in Z-direction with a step size of 1 μm (~8–9 Z-slices) to collect light from the planes above and below the focal plane. The image analysis was performed using FIJI software and in-house written Python code. The MAP2 channel was used to define the region of interest (ROI) of the neurons. The mean fluorescent intensity (total intensity normalized to the area) was measured for the FUNCAT and MAP2 channels for the defined ROIs. The maximum intensity projection of the slices was used for quantification of the mean fluorescent intensities. The mean fluorescent intensity of the FUNCAT channel was normalized to the MAP2 channel for comparison between different treatment conditions.

### Immunostaining for p-eEF2 and p-RPS6

Primary neuronal cultures were treated with nifedipine (50 μM), dantrolene (60 μM), or 2-APB (50 μM) for 10 min, followed by NMDAR or mGluR stimulation (20 μM NMDA or 50 μM DHPG) for 5 min, and fixed with 4% PFA for 10 min (refer to Table 1 for details of the drugs). After 10 min, the neurons were washed thrice with 1X PBS, followed by permeabilization with 0.3% Triton X-100 (made in TBS$_{50}$) for 10 min. This was followed by 1 h of blocking with 2% BSA and 2% FBS prepared in TBS$_{50}$T (with 0.1% Triton X-100). They were incubated with the primary antibody (prepared in blocking buffer) overnight at 4°C. Table 2 contains the details of dilutions and catalog numbers of all the antibodies used. This was followed by three washes with TSB$_{50}$T and 1-h incubation with the secondary antibody (prepared in blocking buffer) at RT. Table 2 contains the details of dilutions and catalog numbers of the secondary antibodies used. After three washes with TBS$_{50}$T, the neurons were mounted using Mowiol. The slides were imaged on an Olympus FV3000 confocal laser scanning upright microscope with a 60X objective. The pinhole was kept at 1 Airy Unit and the optical zoom at 2X to satisfy Nyquist's sampling criteria in the XY-direction. The objective was moved in Z-direction with a step size of 1 μm (~8–9 Z-slices) to collect light from the planes above and below the focal plane. The image analysis was performed using FIJI software. The MAP2 channel was used to define the ROI of the neurons. The maximum intensity projection of the slices was used for quantification of the mean fluorescent intensities. The mean fluorescent intensity of the p-eEF2 or p-RPS6

**Table 2.  Antibodies used for immunostaining.**

| Protein | Dilution | Catalog number, company |
| --- | --- | --- |
| MAP2 | 1:500 | M9942; Sigma-Aldrich |
| MAP2(for puromycin assay) | 1:500 | ab32454; Abcam |
| p-eEF2 | 1:500 | 2331S; Cell Signaling Technologies |
| p-RPS6 | 1:500 | 2217S; Cell Signaling Technologies |
| Puromycin | 1:1,000 | MABE343; Merck Millipore |
| Alexa Fluor 488 | 1:500 | A-11059; Thermo Fisher Scientific |
| Alexa Fluor 555 | 1:500 | A-21428; Thermo Fisher Scientific |

channel was normalized to the MAP2 channel for comparison between different treatment conditions.

### Puromycin incorporation assay

DIV 15 rat primary cortical neurons were subjected to puromycin incorporation under three different conditions–conditioned Neurobasal media (no medium change), media changed to fresh Neurobasal, and media changed to methionine-free DMEM. The different media were incubated for 30 min, followed by treatment with puromycin (5 $\mu$M) for 10 min. After this, the neurons were subjected to the standard immunostaining protocol with overnight incubation of puromycin and MAP2 antibodies. This was followed by secondary antibody incubation for 3 h. Table 2 contains the details of the dilutions and catalog numbers of the antibodies used. The imaging and analysis were performed using the same conditions as described for p-eEF2 and p-RPS6 imaging.

### Treatment of neurons and lysis

Primary neuronal cultures were subjected to NMDAR or mGluR stimulation (20 $\mu$M NMDA or 50 $\mu$M DHPG, respectively) for 1, 5, and 20 min in the presence or absence of the drugs. The neurons were pre-treated with the drugs nifedipine (50 $\mu$M), dantrolene (60 $\mu$M), 2-APB (50 $\mu$M), or MK801 (50 $\mu$M) for 10 min before the stimulation (refer to Table 1 for details of the drugs). In the experiments with no calcium in the media, the neurons were stimulated with NMDA in artificial cerebrospinal fluid (ACSF—120 mM NaCl, 3 mM KCl, 1 mM MgCl$_2$, 3 mM NaHCO$_3$, 1.25 mM NaH$_2$PO$_4$, 15 mM Hepes, 30 mM glucose, pH 7.4) with or without calcium (2 mM CaCl$_2$). For stimulation of L-VGCCs, BayK8644 (100 $\mu$M) was used for 5 and 20 min.

**Table 3.  Antibodies used for Western blotting.**

| Protein | Dilution | Catalog number, company |
| --- | --- | --- |
| eEF2 | 1:1,000 | 2332S; Cell Signaling Technologies |
| p-eEF2 | 1:2,000 | 2331S; Cell Signaling Technologies |
| RPS6 | 1:1,000 | 2211S; Cell Signaling Technologies |
| p-RPS6 | 1:1,000 | 2217S; Cell Signaling Technologies |
| Tuj1 | 1:4,000 | T8578; Sigma-Aldrich |
| Secondary rabbit HRP | 1:5,000 | A0545; Sigma-Aldrich |
| Secondary mouse HRP | 1:5,000 | 31430; Thermo Fisher Scientific |

After the treatment, the cells were lysed in a buffer containing 20 mM Tris–HCl, 100 mM KCl, 5 mM MgCl$_2$, 1% Nonidet P-40 (NP-40), 1 mM dTT, 1X Protease Inhibitor Cocktail, RNase inhibitor, and 1X Phosphatase Inhibitor and centrifuged at 20,000$g$ and 4°C for 20 min. The supernatant was denatured in SDS dye for Western blotting.

### SDS–PAGE and Western blotting

For Western blotting analysis, the denatured lysates were run on 10% resolving and 5% stacking acrylamide gels and subjected to overnight transfer onto the PVDF membrane. The blots were subjected to blocking for 1 h at RT using 5% BSA prepared in TBST (TBS with 0.1% Tween-20). This was followed by primary antibody (prepared in blocking buffer) incubation for 2–3 h at RT. HRP-tagged secondary antibodies were used for primary antibody detection. The blots were incubated with the secondary antibodies (prepared in blocking buffer) for 1 h at RT. Three washes of 5–10 min each were given after primary and secondary antibody incubation using TBST solution. The blots were subjected to chemiluminescent-based detection of the HRP-tagged proteins. Table 3 contains the details of dilutions and catalog numbers of all the antibodies used. For the analysis of eEF2 phosphorylation and RPS6 phosphorylation, the samples were run in duplicates where one set was used to probe for the phospho-proteins (p-eEF2 and p-RPS6), and the other set was used to probe for the total proteins (eEF2 and RPS6). In each set, Tuj1 was used as the loading control. In every set, the blots were cut to probe for eEF2 and RPS6 on the same blot. Hence, the Tuj1 used to normalize eEF2 and RPS6 was the same in a given set. Similarly, p-eEF2 and p-RPS6 were probed on the same blot for a given set, using the corresponding Tuj1 to normalize the given set. All the Western blot quantifications were performed using densitometric analysis on ImageJ software.

### Calcium imaging

Calcium imaging was done on DIV15 neurons plated on 35-mm dishes at a density of 200,000–250,000 cells per dish. The imaging and washes were performed with ACSF media (120 mM NaCl, 3 mM KCl, 1 mM MgCl$_2$, 3 mM NaHCO$_3$, 1.25 mM NaH$_2$PO$_4$, 15 mM Hepes, 2 mM CaCl$_2$, 30 mM glucose, pH 7.4). The cells were washed once with ACSF and incubated at 37°C with 1 ml of freshly prepared Fluo-4 AM dye solution (2 $\mu$M Fluo-4 AM and 0.002% pluronic acid in ACSF) (F14217; Thermo Fisher Scientific) for 20 min. They were given two washes

and incubated in ACSF at 37°C for 10–20 min before imaging. The drugs nifedipine or dantrolene were added during this incubation step (refer to Table 1 for details of the drugs). The neurons were imaged using an Olympus FV3000 confocal laser scanning upright microscope with a 20X water dipping objective, excited with 488-nm lasers. The neurons were imaged for a total of 7 min at a rate of 3 s per frame (140 frames in total). They were imaged in the basal condition for 1 min (20 frames). After that, they were imaged for 5 min (100 frames) on NMDA addition. Finally, they were imaged for 1 min (20 frames) on the addition of ionomycin solution (10 $\mu$M ionomycin with 10 mM CaCl$_2$) (407950; Sigma-Aldrich). The images obtained were analyzed using the Time-Series Analyzer plug-in on FIJI. Only the ionomycin-responsive neurons were selected for the analysis. Hence, the ionomycin frames were used to define the ROIs of the neurons for each series. The average intensities were obtained for the selected ROIs in each frame. The change in fluorescent intensity at each frame was normalized to the fluorescent intensity of the first frame (F0) for each ROI. The normalized change in fluorescent intensity ($\Delta F/F0$) was plotted along the time axis and used for statistical analysis as well.

### Statistical analyses

All statistical analyses were performed using GraphPad Prism software. The normality of the data was checked using the Kolmogorov–Smirnov test. For experiments with less than five data points, parametric statistical tests were applied. Data were represented as the mean ± SEM in all biochemical experiment graphs. FUNCAT and imaging data were represented as boxes and whiskers with all the individual data points. Statistical significance was calculated using an unpaired $t$ test (two-tailed with equal variance) in cases where two groups were being compared. One-way or two-way ANOVA was used for multiple group comparisons, followed by the required multiple comparison test. A $P$-value less than 0.05 was considered to be statistically significant.

## Supplementary Information

## Acknowledgements

This work was primarily supported by the core funds from the Centre for Brain Research (CBR) and the CBR-FABRIC grant. RS Muddashetty, S Ramakrishna and RP Kommaddi thank the funding from CBR. KK Freude was supported by the Novo Nordisk Foundation (GliAD—NNF1818OC0052369 & RhoAD—NNF21OC0071571). BK Radhakrishna received a research fellowship from University Grants Commission (NTA Ref number—191620053012), Government of India. NM Shah received a DST/INSPIRE fellowship (IF200172), Government of India. N Basavaraju received a research fellowship from the Council of Scientific and Industrial Research (CSIR) (File No: 09/1233(13189)/ 2022-EMR-I), Government of India. We thank all the central facilities at CBR-IISc, especially the CAF (Central Animal Facility) at IISc, Liquid Nitrogen Facility at IISc, imaging facility at CBR, primary cell culture facility at CBR, and stem cell culture facility at CBR. We thank Prof. Dasaradhi Palakodeti from the Institute for Stem Cell Science and Regenerative Medicine (inStem), India, for providing the puromycin antibody. We thank Prof. Gaiti Hassan from the National Centre for Biological Sciences (NCBS), India, for her valuable suggestions.

## Author Contributions

S Ramakrishna: conceptualization, data curation, formal analysis, validation, investigation, visualization, methodology, project administration, and writing—original draft, review, and editing.
BK Radhakrishna: conceptualization, formal analysis, validation, investigation, and methodology.
AP Kaladiyil: conceptualization, software, formal analysis, and investigation.
NM Shah: formal analysis, validation, investigation, and methodology.
N Basavaraju: resources and methodology.
KK Freude: resources, methodology, and writing—review and editing.
RP Kommaddi: resources, methodology, and writing—review and editing.
RS Muddashetty: conceptualization, resources, data curation, supervision, funding acquisition, investigation, project administration, and writing—review and editing.

## Conflict of Interest Statement

The authors declare that they have no conflict of interest.

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
