## [Reviewer comments · Life Science Alliance]

Life Science Alliance

Distinct calcium sources regulate temporal profiles of NMDAR and mGluR mediated protein synthesis

Sarayu Ramakrishna, Bindushree Radhakrishna, Ahamed Kaladiyil, Nisa Shah, Nimisha Basavaraju, Kristine Freude, Reddy Kommaddi, and Ravi Muddashetty

DOI: <https://doi.org/10.26508/lsa.202402594>

Corresponding author(s): Ravi Muddashetty, Indian Institute of Science Bangalore

Review Timeline:

Submission Date:	2024-01-14
Editorial Decision:	2024-02-14
Revision Received:	2024-04-22
Editorial Decision:	2024-05-01
Revision Received:	2024-05-02
Accepted:	2024-05-02

Transaction Report:

February 14, 2024

Re: Life Science Alliance manuscript #LSA-2024-02594-T

Prof. Ravi S Muddashetty
Indian Institute of Science Bangalore
Centre for Brain Research
CV Raman Avenue
Bengaluru, Karnataka 560012
India

Dear Dr. Muddashetty,

Thank you for submitting your manuscript entitled "Distinct calcium sources regulate the temporal profile of NMDAR and mGluR mediated protein synthesis in neurons" to Life Science Alliance. The manuscript was assessed by expert reviewers, whose comments are appended to this letter. We invite you to submit a revised manuscript addressing the Reviewer comments.

Thank you for this interesting contribution to Life Science Alliance. We are looking forward to receiving your revised manuscript.

Sincerely,

B. MANUSCRIPT ORGANIZATION AND FORMATTING:

Reviewer #1 (Comments to the Authors (Required)):

The research article by Ramakrishna et al. used primary neurons and iPSC-derived neurons to study the mechanisms of NMDAR- and mGluR1/5-mediated protein synthesis. The topic is directly related to receptor function in plasticity in health and disease (i.e., Alzheimer's in this study). The data collection and analysis are rigorous and sufficient. The approaches involved multi-level investigation with calcium imaging, molecular examination, pharmacological intervention, and direct labeling of new protein synthesis. The conclusion is sufficiently supported by data. The manuscript is well-written and presented in a meaningful way. The outcome, which reveals the function of different calcium sources in regulating the dynamic dimension of activity-dependent protein synthesis, provides new insights into the cellular aspects of neuroplasticity in both health and disease. I endorse the publication.

Reviewer #2 (Comments to the Authors (Required)):

Ramakrishna et al.

Distinct calcium sources regulate the temporal profiles of NMDAR and mGluR mediated protein synthesis in neurons

In this manuscript, Ramakrishna et al. demonstrate a set of experiments ultimately identifying IP3R and SOCE as crucial mediators of both the early and late phases of mGluR-mediated protein synthesis, and further identify L-VGCC, SOCE, and RyR as downstream mediators of NMDAR-dependent protein synthesis. Specifically, the authors expand on previous work of theirs showing immediate NMDAR-dependent translation inhibition in response to NMDA— here identifying a delayed L-VGCC-mediated switch to translation activation. The authors expand on the relevance of these findings in two models of Alzheimer's Disease (AD) - APP/PS1 double transgenic AD mice and PSEN1 L150P mutant iPSC's confirming impairments in Ca²⁺ signalling and translation. Overall, the experiments presented are thorough and address the aims of the manuscript, figures are presented clearly (with minor comments), and experimental samples sizes and statistical analyses appear suitable. Additionally, this works gains strength through the addition of AD models. However, prior to publication, one major comment should be addressed (below) to verify their claims.

Major

- It is noted that the authors used methionine-free DMEM during the incorporation of AHA for FUNCAT experiments, as opposed to methionine-free Neurobasal medium. Use of DMEM is concerning and is a likely source of stress for the neurons (evidence of this in non-uniform MAP2 staining, dendrite thinning, and occasional blebbing in Figs. 1H-I, Fig. 5C) compared to robust MAP2 staining in samples processed for immunostaining of p-eEF2 and p-RPS6 (Fig. 5F-G) understood to have been previously cultured (or drug-treated) in Neurobasal medium before fixation. Cellular stress (nutrient deprivation, glucose stress, proteasomal inhibition) all can lead to changes in protein synthesis output- given the physiological differences between DMEM and neurobasal- this alone may result in sufficient cellular stress which could result in the change in protein synthesis output the authors describe. The authors should confirm normal physiology (firing rates etc) is maintained during this period of DMEM, and observe if DMEM alone changes protein synthesis output (using puromycin rather than AHA which would not require methionine deprivation).
- In mGluR experiments, the authors state: "FUNCAT signal was not normalized to MAP2 in these experiments as we observed an increase in the MAP2 levels on 5 and 20 minutes of mGluR stimulation (Fig S4A)". From Supplementary Figure 4 it is understood this is an increase in MAP2 intensity, have the authors considered normalising FUNCAT signal to MAP2 area? Is there any evidence that mGluR stimulation increases MAP2 synthesis? Clarify whether DHPG treatment increased MAP2 area. In all other instances (FUNCAT, immunocytochemistry) define whether normalisation was to MAP2 area or intensity.

Minor

- Figure 1H-J, Figure 5F&G it is recommended to keep consistency between Y-axes across graphs.
- Figures 2B & 2D the authors state "At 5-minute NMDAR stimulation, though eEF2 phosphorylation was lower than 1-minute, it was still higher than untreated conditions indicating a net translation inhibition (Fig 2B, 2D...)," ensure statistics for this claim is clear.

- Define the neuronal compartment(s) included in ROI analysis for both calcium and immunofluorescent experiments. Define how this ROI was created (i.e. MAP2 mask, segmentation)
- Current size of all scale bars renders them illegible across all figures.
- IPSCs

Reviewer #3 (Comments to the Authors (Required)):

In this manuscript by Ramakrishna et. al., the authors build up on their previous findings where they had observed that neurons exposed to APOE4 exhibit a suppression of both basal and NMDAR-mediated protein synthesis responses. In this study the authors demonstrate how calcium responses generated by different sources differentially modulate neuronal activity-mediated protein synthesis. Specifically, they show that stimulation of NMDARs generates a bi-phasic protein synthesis response where the neuronal protein synthesis is inhibited within 1 minute of NMDAR stimulation, which comes down to basal level around 5 minutes with a significant reduction in protein synthesis around 20 minutes after NMDAR stimulation. Further they present data to show that the NMDAR-mediated translation activation requires the contribution of L-VGCCs, Ryanodine receptors, and SOCE.

The authors show that in contrast to NMDAR, stimulation of mGluR leads to overall neuronal translation upregulation, which is dependent on IP3- and SOCE-mediated calcium release. In the end the authors use human fAD iPSCs derived glutamatergic neurons and APP/PS1 transgenic Alzheimer's disease [AD] mice to conclude that the NMDAR- mediated calcium release and translation response is dysregulated in AD neurons.

Overall, this is a rigorous and an interesting manuscript, which provides new insights about how different calcium sources differentially regulate neuronal protein synthesis. I have few concerns as below, which needs to be taken care of before the manuscript can be formally accepted for publication.

Major Points:

1. In Fig. 2B, the blots are overexposed because of which the quantifications are not convincing. The authors should provide lower exposure images.
2. In Fig. 2D, again the blots are not representative. Also, the loading pattern in 2B is different than all other blots in the manuscript.
3. In Fig. 2E, the IF data does not match with the western blot quantifications. The authors need to provide explanation for this discrepancy.
4. In Fig. 3B-E, like above the IF data and WB data do not match and the main text lacks any explanation.
5. In Fig. 3F, I was expecting to see a reduction in the phospho-RPS6 level at the 1 minute time point compared to no treatment, why there no reduction? Is it possible that, during NMDAR stimulation the initial reduction in protein synthesis is primarily due to inhibition in translation elongation but at later time points increase in translation initiation drives the upregulation of protein synthesis?
6. The rationale for checking neurodegeneration is lacking in the manuscript.

Response to Reviewers' comments

Reviewer 1

The research article by Ramakrishna et al. used primary neurons and iPSC-derived neurons to study the mechanisms of NMDAR- and mGluR1/5-mediated protein synthesis. The topic is directly related to receptor function in plasticity in health and disease (i.e., Alzheimer's in this study). The data collection and analysis are rigorous and sufficient. The approaches involved multi-level investigation with calcium imaging, molecular examination, pharmacological intervention, and direct labeling of new protein synthesis. The conclusion is sufficiently supported by data. The manuscript is well-written and presented in a meaningful way. The outcome, which reveals the function of different calcium sources in regulating the dynamic dimension of activity-dependent protein synthesis, provides new insights into the cellular aspects of neuroplasticity in both health and disease. I endorse the publication.

We are pleased to know that the reviewer appreciates the approaches used in the study, analysis, conclusions and the impact of the findings. We sincerely thank the reviewer for endorsing the publication of the manuscript.

Reviewer 2

In this manuscript, Ramakrishna et al. demonstrate a set of experiments ultimately identifying IP3R and SOCE as crucial mediators of both the early and late phases of mGluR-mediated protein synthesis, and further identify L-VGCC, SOCE, and RyR as downstream mediators of NMDAR-dependent protein synthesis. Specifically, the authors expand on previous work of theirs showing immediate NMDAR-dependent translation inhibition in response to NMDA⁺— here identifying a delayed L-VGCC-mediated switch to translation activation. The authors expand on the relevance of these findings in two models of Alzheimer's Disease (AD) - APP/PS1 double transgenic AD mice and PSEN1 L150P mutant iPSC's confirming impairments in Ca²⁺ signalling and translation. Overall, the experiments presented are thorough and address the aims of the manuscript, figures are presented clearly (with minor comments), and experimental samples sizes and statistical analyses appear suitable. Additionally, this work gains strength through the addition of AD models. However, prior to publication, one major comment should be addressed (below) to verify their claims.

We thank the reviewer for the appreciation of the experimental methods and statistical analyses presented in the manuscript. We thank the reviewer for the critical comments. We have tried our best to address all the major and minor concerns. The list of the changes made in the figures and the text is included at the end of this document. The response to the comments is as follows –

Major comments

1. It is noted that the authors used methionine-free DMEM during the incorporation of AHA for FUNCAT experiments, as opposed to methionine-free Neurobasal medium. Use of DMEM is concerning and is a likely source of stress for the neurons (evidence of this in non-uniform MAP2 staining, dendrite thinning, and occasional blebbing in Figs. 1H-I, Fig. 5C) compared to robust MAP2 staining in samples processed for immunostaining of p-eEF2 and p-RPS6 (Fig. 5F-G) understood to have been previously cultured (or drug-treated) in Neurobasal medium before fixation. Cellular stress (nutrient deprivation, glucose stress, proteasomal inhibition) all can lead to changes in protein synthesis output- given the physiological differences between DMEM and neurobasal- this alone may result in sufficient cellular stress which could result in the change in protein synthesis output the authors describe. The authors should confirm normal physiology (firing rates etc) is maintained during this period of DMEM, and observe if DMEM

alone changes protein synthesis output (using puromycin rather than AHA which would not require methionine deprivation).

We understand the concern of the reviewer with respect to the use of Methionine-free DMEM instead of Neurobasal. Methionine-free Neurobasal is not easily available and hence all the standard FUNCAT protocols use Met-free DMEM instead. As the reviewer has suggested, we have performed Puromycin incorporation assay to measure the effect of changing the media to Met-free DMEM on neuronal protein synthesis. To do this, we used three conditions - i) We did the puromycin incorporation in conditioned neurobasal without changing the media, ii) We changed the media from neurobasal to Met-free DMEM before puromycin incorporation and iii) We changed the media to fresh neurobasal before puromycin incorporation. There was no significant change in the puromycin incorporation in all three conditions. Hence, incubation of neurons with Met-free DMEM alone does not cause any significant changes in protein synthesis compared to incubation with fresh neurobasal or conditioned neurobasal (no media change). These results are included in Supplementary Figure 1 (Fig S1D). As a control for the puromycin incorporation assay, we treated the neurons with Anisomycin ($5\mu\text{M}$, 35 minutes) to verify that the signal was protein synthesis dependent. Puromycin incorporation was completely abolished on Anisomycin treatment. The Puromycin and MAP2 staining data on Anisomycin treatment is shown below but not included in the manuscript. The MAP2 staining looks uniform and unchanged between the 3 conditions. Hence, we think it is unlikely that incubation with Met-free DMEM has caused a stress response in our experiments.

N=20-25 neurons from 2 independent experiments. One-way ANOVA followed by Tukey's multiple comparison test

We went through all the MAP2 images corresponding to the FUNCAT experiments in the manuscript, and we do not observe blebbing or thinning in majority of the cases. We apologize that some of the representative images we had used were not good. Accordingly, we have changed the representative images for Fig 1H (untreated condition, NMDA 20'), Fig 1I (Nifi+NMDA 20'), Fig 1J (Dant+NMDA 20'), Fig 5C (WT neurons NMDA 5' and NMDA 20'). We are happy to upload the raw images of FUNCAT/MAP2 for all the conditions. We thank the reviewer for this suggestion as we believe that additional validation through the puromycin assay has made our data stronger.

2. In mGluR experiments, the authors state: "FUNCAT signal was not normalized to MAP2 in these experiments as we observed an increase in the MAP2 levels on 5 and 20 minutes of mGluR stimulation (Fig S4A)". From Supplementary Figure 4 it is understood this is an increase in MAP2 intensity, have the authors considered normalising FUNCAT signal to MAP2 area? Is there any evidence that mGluR stimulation increases MAP2 synthesis? Clarify whether DHPG treatment increased MAP2 area. In all other instances (FUNCAT, immunocytochemistry) define whether normalisation was to MAP2 area or intensity.

In all the imaging experiments, we have used MAP2 channel to define the ROI. For each ROI, we have quantified the mean intensity for the desired channel (FUNCAT/p-eEF2/p-RPS6) and MAP2 channel. Mean intensity is calculated as total intensity normalized to the area. Further, we have normalized the mean intensity of the desired channel (FUNCAT/p-eEF2/p-RPS6) to the mean intensity of the MAP2 channel. Hence, the area quantified would be the same for the desired channels and the MAP2 channel, and it is taken into consideration in our quantifications. In case of mGluR stimulation, though the FUNCAT and MAP2 graphs are presented separately, the ROIs for quantification are defined using MAP2 channel and the area quantified would be the same for the FUNCAT and MAP2 channels. We do not observe an increase in MAP2 area on mGluR stimulation, it is only an increase in intensity. We thank the reviewer for this comment as we realized that we had not described the image analysis and area normalizations clearly. We have now elaborated the image analysis protocol in the material/methods section.

Further, we found another study (*Charlotte A. G.H. van Gelder et al, 2020; Temporal Quantitative Proteomics of mGluR-induced Protein Translation and Phosphorylation in Neurons*) reporting the increase of MAP2 protein levels on mGluR stimulation in neurons. The study performed proteomics analysis to identify the proteins translated on mGluR stimulation and found that several microtubule-associated proteins (Map2, Map6, Mapre1, Map11c3b, and Mapt) were part of this list. We have now included this reference in the manuscript.

Minor comments

1. Figure 1H-J, Figure 5F&G it is recommended to keep consistency between Y-axes across graphs.

The y-axis scale has been changed to keep it consistent in Figure 1H-J and Figure 5F-G.

2. Figures 2B & 2D the authors state "At 5-minute NMDAR stimulation, though eEF2 phosphorylation was lower than 1-minute, it was still higher than untreated conditions indicating a net translation inhibition (Fig 2B, 2D...)," ensure statistics for this claim is clear.

We made this claim since we had reported this temporal profile of eEF2 phosphorylation in our previous study (*Ramakrishna S et al, 2021; APOE4 affects basal and NMDAR-mediated protein synthesis in neurons by perturbing calcium homeostasis*) and we observed the same pattern in the current study as well. However, we agree with the reviewer that our statistical analysis in the current study (Figure 2B, 2D) does not compare eEF2 phosphorylation within the NMDAR stimulation time points and limits us from making this claim. Hence, we have now changed the statement and provided the reference from our previous study to support the claim. It is now written as – “As observed in our previous study, at 5-minute NMDAR stimulation, eEF2 phosphorylation seemed lower than 1-minute and higher than untreated conditions indicating a net translation inhibition.”

3. Define the neuronal compartment(s) included in ROI analysis for both calcium and immunofluorescent experiments. Define how this ROI was created (i.e. MAP2 mask, segmentation)

We thank the reviewer for bringing this to our notice. We have defined the basis for ROI creation for FUNCAT, immunostaining and calcium imaging analyses in the material and method sections. In case of FUNCAT and immunostaining experiments, MAP2 channel was used to define the ROI of the neurons. The mean fluorescent intensity (total intensity normalized to the area) was measured for the desired channel (FUNCAT/Puromycin/p-eEF2/p-RPS6) and MAP2 channel for the defined ROIs. In case of calcium imaging experiments, the ionomycin

responsive neurons were selected for the analysis. Hence, the ionomycin frames were used to define the ROIs of the neurons, whose average intensities were measured for the entire time series.

4. **Current size of all scale bars renders them illegible across all figures.**

The thickness of the scale bar has been increased for all the images and the units for the scale bar is included in the figure legends.

Reviewer 3

In this manuscript by Ramakrishna et. al., the authors build up on their previous findings where they had observed that neurons exposed to APOE4 exhibit a suppression of both basal and NMDAR-mediated protein synthesis responses. In this study the authors demonstrate how calcium responses generated by different sources differentially modulate neuronal activity-mediated protein synthesis. Specifically, they show that stimulation of NMDARs generates a bi-phasic protein synthesis response where the neuronal protein synthesis is inhibited within 1 minute of NMDAR stimulation, which comes down to basal level around 5 minutes with a significant reduction in protein synthesis around 20 minutes after NMDAR stimulation. Further they present data to show that the NMDAR-mediated translation activation requires the contribution of L-VGCCs, Ryanodine receptors, and SOCE. The authors show that in contrast to NMDAR, stimulation of mGluR leads to overall neuronal translation upregulation, which is dependent on IP3- and SOCE-mediated calcium release. In the end the authors use human fAD iPSCs derived glutamatergic neurons and APP/PS1 transgenic Alzheimer's disease [AD] mice to conclude that the NMDAR- mediated calcium release and translation response is dysregulated in AD neurons. Overall, this is a rigorous and an interesting manuscript, which provides new insights about how different calcium sources differentially regulate neuronal protein synthesis. I have few concerns as below, which needs to be taken care of before the manuscript can be formally accepted for publication.

We sincerely thank the reviewer for finding the concept interesting and the work rigorous. We thank the reviewer for all the comments and suggestions. We have tried our best to address the concerns of the reviewer to make the manuscript suitable for publication. The list of the changes made in the figures and the text is included at the end of this document. The response to the comments is as follows –

1. **In Fig. 2B, the blots are overexposed because of which the quantifications are not convincing. The authors should provide lower exposure images.**

We have changed the representative blot in Fig 2B to provide a lower exposure image in the same loading pattern as others.

2. **In Fig. 2D, again the blots are not representative. Also, the loading pattern in 2B is different than all other blots in the manuscript.**

We have changed the representative blot for Fig 2D.

3. **In Fig. 2E, the IF data does not match with the western blot quantifications. The authors need to provide explanation for this discrepancy.**

We think that the differences we observed in western blotting and IF data could be due to the following reasons – 1. Western blotting is processed through batch treatment and lysis, while we pick MAP2 positive neurons in case of immunostaining and imaging. 2. Tuj1 is used as the normalizing factor for western blotting while we use MAP2 in imaging experiments. We have now highlighted this discrepancy and provided the plausible explanation in the discussion section of the manuscript.

4. In Fig. 3B-E, like above the IF data and WB data do not match and the main text lacks any explanation.

As mentioned in the previous comment, we have now provided a plausible explanation for this discrepancy in the discussion section of the manuscript. We think that the differences we observe in western blotting and IF data could be due to the following reasons – 1. Western blotting is processed through batch treatment and lysis, while we pick MAP2 positive neurons in case of immunostaining and imaging. 2. Tuj1 is used as the normalizing factor for western blotting while we use MAP2 in imaging experiments.

5. In Fig. 3F, I was expecting to see a reduction in the phospho-RPS6 level at the 1 minute time point compared to no treatment, why there no reduction? Is it possible that, during NMDAR stimulation the initial reduction in protein synthesis is primarily due to inhibition in translation elongation but at later time points increase in translation initiation drives the upregulation of protein synthesis

We agree with the reviewer. We also think that the initial reduction of protein synthesis is primarily mediated by translation elongation whereas the later time points of translation activation have contribution by both elongation and initiation steps. We think that the initial halt on translation elongation accompanied by degradation of existing proteins, followed by increased translation initiation and elongation could lead to the change of overall translome/proteome on NMDAR stimulation. This is a very interesting thought which we had not discussed well in the manuscript. We thank the reviewer for pointing this out and we have now included this in the discussion section.

6. The rationale for checking neurodegeneration is lacking in the manuscript.

We thank the reviewer for bringing this to our notice. We have now included the explanations for investigating our findings in the AD model in results and discussion sections. The explanation is - Disruption of calcium homeostasis is an important early phenotype in AD. AD neurons are shown to have defective calcium signaling involving multiple channels and sensors like NMDARs, L-VGCCs, STIM1, RyRs, and SOCE^{33,34,36,71}. Specific calcium blockers against these channels are shown to protect the neurons from A β toxicity as well. Parallely, many studies have reported impaired protein synthesis in AD, including the dysregulation of translation initiation and elongation factors^{44,73-76}. Yet, the defect in activity-mediated translation or the link between calcium and translation defect in AD is not investigated. In this context, our results demonstrating the NMDAR-mediated calcium and translation defect in AD neurons would help in explaining the dysregulated synaptic signaling and plasticity in AD.

List of changes made in the manuscript

Author order and name

Ahamed Panikkaveettil Kaladiyil changed from 4th author to 3rd author

Nisa Manzoor Shah changed from 3rd author to 4th author

Reddy P Kommaddi changed to Reddy Peera Kommaddi

Email addresses

Sarayu Ramakrishna – sarayur30@gmail.com

Bindushree K Radhakrishna – bindushreekr@cbr-iisc.ac.in

Nisa Manzoor Shah – shahnisa@cbr-iisc.ac.in

Nimisha Basavaraj – nimishab@cbr-iisc.ac.in

Reddy Peera Kommaddi – reddy@cbr-iisc.ac.in

Ravi S Muddashetty – ravimshetty@cbr-iisc.ac.in

Figures

Figure number	Changes made
Figure 1H	Scale bar thickness increased for all images Representative image for Untreated and NMDA 20' condition is changed
Figure 1I	Scale bar thickness increased for all images Y-axis of graph changed (increased from 3 to 4) Representative image for Nifi+NMDA 20' condition is changed
Figure 1J	Scale bar thickness increased for all images Y-axis of graph changed (increased from 3 to 4) Representative image for Dant+NMDA 20' condition is changed
Figure 2B	Representative blot changed
Figure 2C	Scale bar thickness increased for all images
Figure 2D	Representative blot changed
Figure 2E	Scale bar thickness increased for all images
Figure 3C	Scale bar thickness increased for all images
Figure 3E	Scale bar thickness increased for all images
Figure 4D	Scale bar thickness increased for all images
Figure 3E	Scale bar thickness increased for all images
Figure 5C	Scale bar thickness increased for all images Representative image for NMDA 5' in WT neurons is changed Representative image for NMDA 20' in WT neurons is changed
Figure 5G	Y-axis of graph changed (increased to 4)

Supplementary Figures

Figure number	Changes made
Figure S1D	New figure
Figure S1E	Previously S1D, figure number changed due to addition of new data
Figure S1F	Previously S1E, figure number changed due to addition of new data

Main text

Section	Changes corresponding to -	Reviewer addressed	comment
---------	----------------------------	--------------------	---------

Results	Figure S1D – puromycin incorporation assay to validate the effect of Met-free DMEM on translation	Reviewer 2 (major comment 1)
	Figure 2B, 2D – eEF2 phosphorylation profile on NMDAR stimulation	Reviewer 2 (minor comment 2)
	Figure S4A – inclusion of reference for MAP2 increase on mGluR stimulation	Reviewer 2 (major comment 2)
	Figure 5 – rationale for checking neurodegeneration	Reviewer 3 (comment 6)
Discussion	Contribution by translation elongation and initiation on NMDAR stimulation	Reviewer 3 (comment 5)
	Discrepancy in western blotting and immunostaining data	Reviewer 3 (comment 3,4)
Material and method	FUNCAT – ROI definition and area normalization for image analysis	Reviewer 2 (major comment 2, minor comment 4)
	Immunostaining - ROI definition for image analysis	Reviewer 2 (minor comment 4)
	Calcium imaging - ROI definition for image analysis	Reviewer 2 (minor comment 4)
	Puromycin incorporation assay – added newly in revision	Reviewer 2 (major comment 1)
	Table 2 – added details of Puromycin and corresponding MAP2 antibody	

May 1, 2024

RE: Life Science Alliance Manuscript #LSA-2024-02594-TR

Prof. Ravi S Muddashetty
Indian Institute of Science Bangalore
Centre for Brain Research
CV Raman Avenue
Bengaluru, Karnataka 560012
India

Dear Dr. Muddashetty,

Thank you for submitting your revised manuscript entitled "Distinct calcium sources regulate the temporal profile of NMDAR and mGluR mediated protein synthesis". We would be happy to publish your paper in Life Science Alliance pending final revisions necessary to meet our formatting guidelines.

- please be sure that the authorship listing and order is correct
- please upload your main manuscript text as an editable doc file
- please add ORCID ID for the corresponding author -- you should have received instructions on how to do so
- please upload all figure files as individual ones, including the supplementary figure files; all figure legends should only appear in the main manuscript file
- please add the Twitter handle of your host institute/organization as well as your own or/and one of the authors in our system
- please note that titles in the system and manuscript file must match
- please upload your Tables in editable .doc or Excel format. They can be included at the bottom of the main manuscript file or sent as separate files.

A. FINAL FILES:

B. MANUSCRIPT ORGANIZATION AND FORMATTING:

Sincerely,

May 2, 2024

RE: Life Science Alliance Manuscript #LSA-2024-02594-TRR

Prof. Ravi S Muddashetty
Indian Institute of Science Bangalore
Centre for Brain Research
CV Raman Avenue
Bengaluru, Karnataka 560012
India

Dear Dr. Muddashetty,

Thank you for submitting your Research Article entitled "Distinct calcium sources regulate temporal profiles of NMDAR and mGluR mediated protein synthesis". It is a pleasure to let you know that your manuscript is now accepted for publication in Life Science Alliance. Congratulations on this interesting work.

DISTRIBUTION OF MATERIALS:

Again, congratulations on a very nice paper. I hope you found the review process to be constructive and are pleased with how the manuscript was handled editorially. We look forward to future exciting submissions from your lab.

Sincerely,
